# The transcription factor Zfh1 acts as a wing-morph switch in planthoppers

Jin-Li Zhang[1], Sun-Jie Chen[1], Xin-Yang Liu[1], Armin P. Moczek[2] & Hai-Jun Xu [1] ✉

Insect wing polyphenism is characterized by its ability to produce two or more distinct wing morphs from a single genotype in response to changing environments. However, the molecular basis of this phenomenon remains poorly understood. Here, we identified a zinc finger homeodomain transcription factor Zfh1 that acts as an upstream regulator for the development of long-winged (LW) or shorted-winged (SW) morphs in planthoppers. Knockdown of *Zfh1* directs SW-destined nymphs to develop into LW morphs by down-regulating the transcriptional level of *FoxO*, a prominent downstream effector of the insulin/IGF signaling (IIS) pathway. The balance between transcriptional regulation via the Zfh1-FoxO cascade and post-translational regulation via the IIS-FoxO cascade provides a flexible regulatory mechanism for the development of alternative wing morphs. These findings help us understand how phenotypic diversity is generated by altering the activity of conserved proteins, and provide an extended framework for the evolution of wing morphological diversity in insects.

The evolution of wings was one of the most important events in the diversification of insects[1,2]. Yet despite the evolutionary advantages of flight, nearly all originally winged insect orders possess many partially winged or secondarily wingless lineages[3–5]. Wing polymorphism is an especially common version of this phenomenon, in which both flying and flightless forms are contained within the same population. Typically, the flying morph possesses fully developed wings and flight muscles, suited for long-distance dispersal. By contrast, the flightless morph possesses vestigial or no wings as well as undeveloped flight muscles, and does not disperse. Winged and wingless morphs maintain high fitness by specializing on different life histories: a winged morph able to disperse when local conditions are declining, and a wingless and more sedentary morph able to invest heavily into reproduction when local conditions are favorable[6,7]. Wing polymorphic insects therefore provide many promising opportunities to investigate the adaptive significance of dispersal polymorphisms and more generally the evolution of alternative life histories. Doing so, however, is hampered by our incomplete understanding of the molecular basis underlying the development of alternative winged and wingless morphs.

Studies in diverse taxa including aphids, crickets, and planthoppers indicate that insect wing polymorphism can be caused by environmental cues encountered during particular developmental stages (polyphenism), by different genotypes, or by a combination of both[7–9]. Juvenile hormone (JH), in particular, has been the main subject of studies into the endocrine regulation of wing polyphenism in diverse insect species, yet persuasive direct evidence documenting a functional role of JH in wing polyphenisms is still lacking[10–12]. In contrast, growing evidence indicates that insect wing polyphenism might be regulated by diverse genes and gene regulatory networks whose identities may depend greatly on the taxa under study: for example, winged or wingless morphs are regulated, at least in part, by ecdysone signaling in female aphids *Acyrthosiphon pisum*[13], whereas the male dimorphism is genetically controlled by a single locus on the X chromosome[14–16]. In another aphid species, *Aphis citricidus*, small RNAs (e.g., miR-9b) also contribute significantly to wing dimorphism[17].

These results contrast to findings in the wing polymorphic brown planthopper (BPH), *Nilaparvata lugens* (Hemiptera: Delphacidae), a destructive rice pest in Asia. During nymphal development, comprised of five nymphal stages, BPH wing buds grow roughly proportionally

[1]State Key Laboratory of Rice Biology, Key Laboratory of Biology of Crop Pathogens and Insects of Zhejiang Province, Institute of Insect Sciences, College of Agriculture and Biotechnology, Zhejiang University, 866 Yu-Hang-Tang Ave, Hangzhou, China. [2]Department of Biology, Indiana University, 915 East 3rd Street, Bloomington, IN 47405, USA. ✉e-mail: haijunxu@zju.edu.cn

with body size. Nymphs eventually mature into distinct, short (SW)- and long-winged (LW) morphs which become externally distinguishable for the first time following the terminal nymphal-to-adult molt. SW adults possess vestigial forewings and rudimentary hindwing buds, in contrast to fully developed forewings and hindwings typical of LW adults (Fig. 1a, Supplementary Fig. 1a). Diverse environmental cues, including crowding, host plant quality, photoperiod, and temperature, have been shown to influence BPH wing-morph switching, although dominant environmental regulators—if they indeed exist—remain yet to be identified[12,18]. The ability to develop into the LW morph enables BPHs to migrate northward from tropical to subtropical areas in the spring, followed by returning migrations in the autumn, causing severe rice yield loss throughout Asia[19–21]. Previous studies showed that the

forkhead transcription factor subgroup O (FoxO) serves as a key regulator that directs the development of LW or SW morphs[22,23], and may play an important role in interfacing with environmental cues such as wounding and stress[24,25]. Additional studies indicated that the insulin/IGF signaling (IIS) pathway affects BPH wing dimorphism via modulating the phosphorylation level of FoxO[22,26,27], resulting in phosphorylated FoxO to be excluded from the nucleus and thus unable to inhibit wing development, thereby leading to LW morphs. Interestingly, the IIS pathway has also been implicated in the regulation of wing dimorphism in both the red-shouldered soapberry bug *Jadera haematoloma*[28] and the European firebug *Pyrrhocoris apterus*[29], likely constituting independent yet parallel co-option events. Yet despite the prominence of the IIS-FoxO signaling axis in the regulation of wing

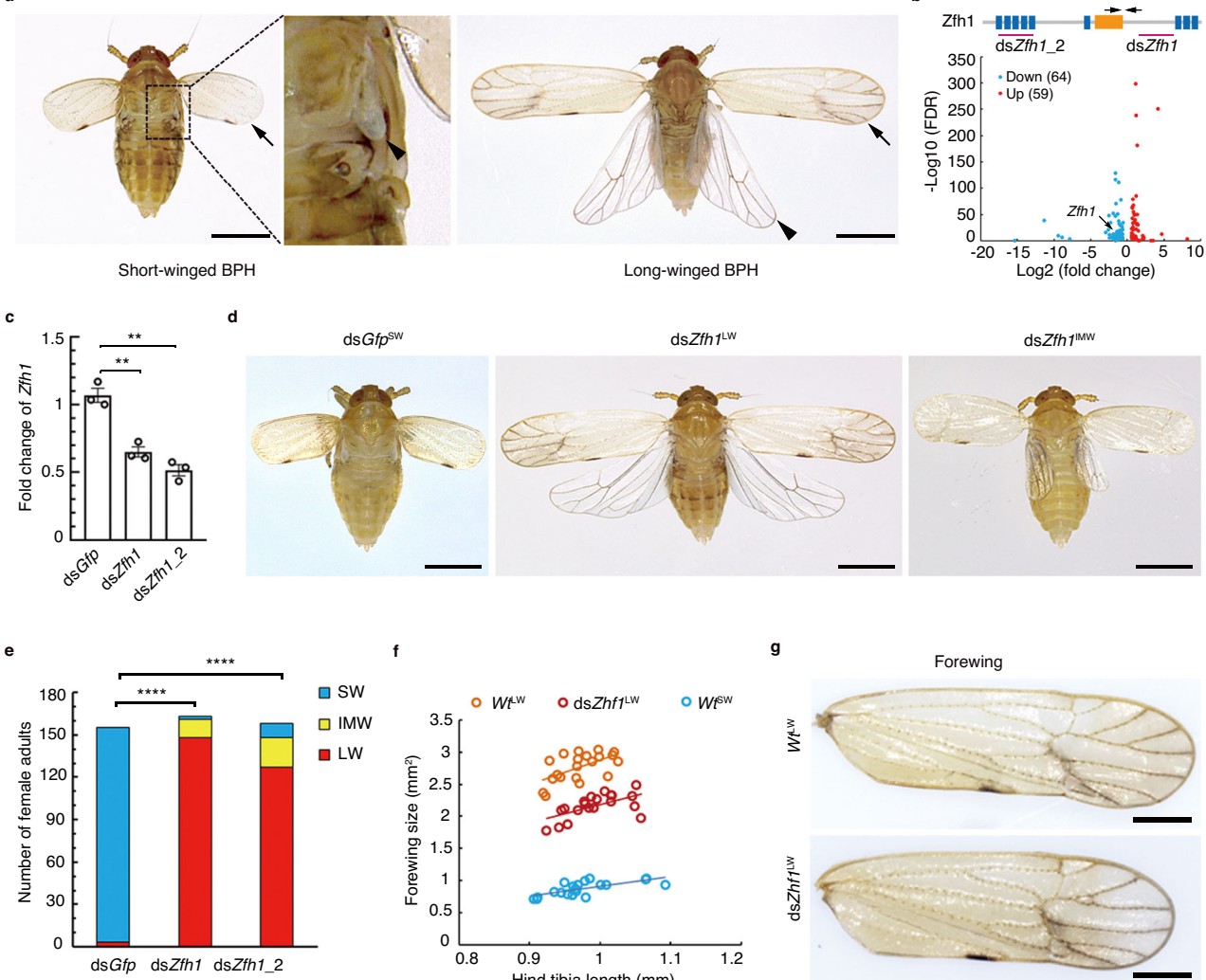

**Fig. 1 | Knockdown of BPH *Zfh1* leads to long-winged morphs. a** Morphology of wild-type long-winged (LW) and short-winged (SW) female adults. Arrows, forewings. Arrowheads, hindwings. **b** Differentially expressed transcription factors between LW-destined ($Wt^{LW}$) and SW-destined ($Wt^{SW}$) fifth-instar nymphs. Thoracic nota were dissected from fifth-instar nymphs for RNA sequencing. Genes differentially expressed between $Wt^{LW}$ and $Wt^{SW}$ with a false discovery rate (FDR) < 0.01 are considered to be significant. The Zfh1 protein contains a homeodomain (orange rectangle) and nine zinc-finger motifs (blue boxes). Regions targeted by dsRNAs (ds*Zfh1* and ds*Zfh1_2*) are underlined. The PCR amplified region used to examine RNAi efficiency is indicated by arrows. **c** RNAi efficiency of ds*Zfh1* and ds*Zfh1_2*. Third-instar $Wt^{SW}$ nymphs were microinjected with ds*Zfh1* or ds*Zfh1_2*. Individuals ($n = 5$) were collected for qRT-PCR assay two days after microinjection. The relative expression of *Zfh1* was normalized to the expression of the ribosomal protein S11 gene (*rps11*).

The experiments were repeated three times with similar results (circles). Data are presented as mean ± SEM. Two-tailed unpaired *t*-test was used for the statistical analysis (\*\* $P = 0.0028$ for ds*Zfh1* vs ds*Gfp* and \*\* $P = 0.0011$ for ds*Zfh1_2* vs ds*Gfp*). **d** Morphology of ds*Gfp*^SW, ds*Zfh1*^LW, and ds*Zfh1*^IMW females. ds*Gfp*^SW, ds*Gfp*-treated BPHs with short wings (SW). ds*Zfh1*^LW and ds*Zfh1*^IMW, ds*Zfh1*-treated BPHs with long and intermediate-size wings (IMW), respectively. Scale bars in (**a**) and (**c**), 1 mm. **e** Number of females with different wing morphs following dsRNA treatments. The LW ratio is compared between two groups using Pearson $\chi^2$ test (\*\*\*\*$P = 1.1716\text{E-}56$, $\chi^2 = 251.586$ and df = 1 for ds*Zfh1* vs ds*Gfp*; \*\*\*\*$P = 4.9769\text{E-}45$, $\chi^2 = 198.2772$ and df = 1 for ds*Zfh1_2* vs ds*Gfp*). **f** Wing size and hind tibia length in ds*Zfh1*^LW, $Wt^{LW}$ and $Wt^{SW}$ females. Each circle represents an individual female ($n = 20$). **g** Vein patterning on forewings from $Wt^{LW}$ and ds*Zfh1*^LW females. Twenty samples were repeated independently with similar results. Scale bars, 500 μm. Source data are provided as a Source Data file.

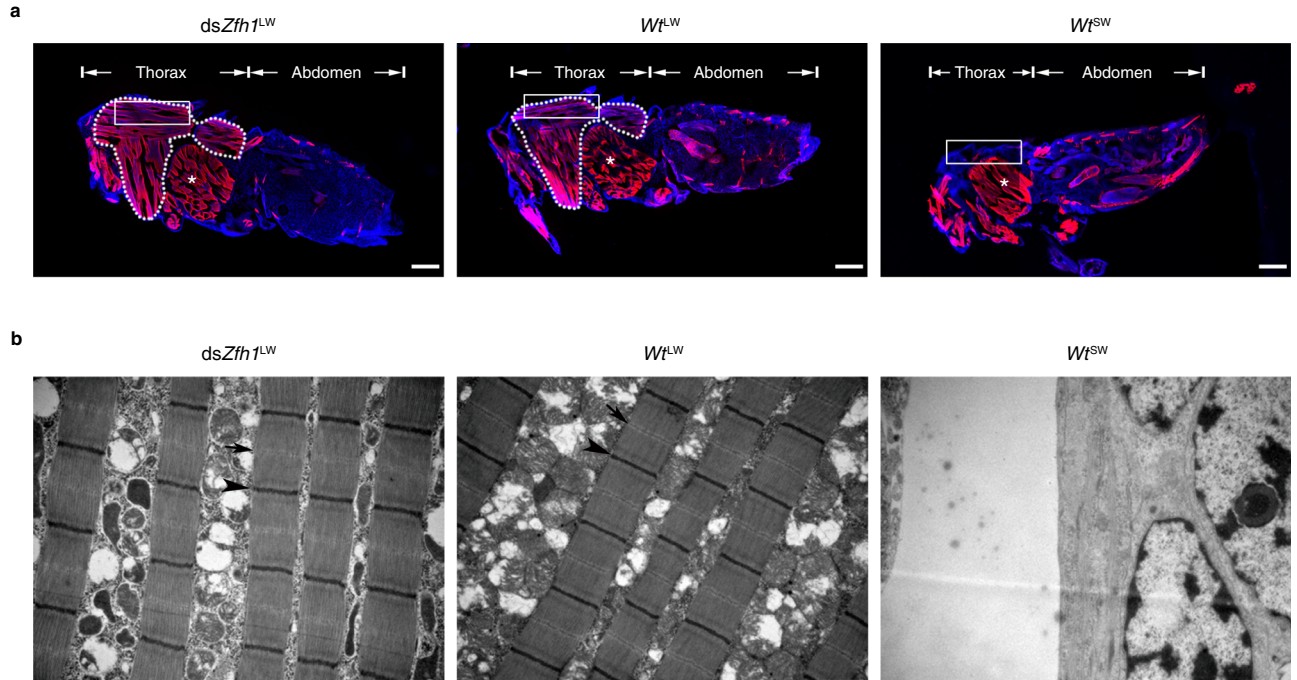

**Fig. 2 | Knockdown of *Zfh1* induces the development of indirect flight muscles. a** Immunohistochemistry staining of indirect flight muscles (IFM). Third-instar *Wt*[SW] nymphs were microinjected with ds*Zfh1*. Thorax and abdomen of emerged female adults were longitudinally cut into sections, followed by staining with rhodamine-labeled phalloidin (red) and DAPI (blue). The IFM is framed by a dotted line, and the trochanteral depressor muscle is indicated by stars. The part of muscles used for transmission electron microscopy is indicated by rectangles. Scale bars, 200 μm. **b** Examination of IFM by transmission electron microscopy. Thoraxes were dissected from female adults and then cut into sections for IFM examination. Arrows and arrowheads represent Z and M discs of sarcomeres, respectively. Scale bars, 1 μm. The experiments in (**a**) and (**b**) were repeated six times independently with similar results. ds*Zfh1*[LW], ds*Zfh1*-treated BPHs with long wings. *Wt*[LW] and *Wt*[SW], wild-type long-winged and short-winged BPHs, respectively.

dimorphism, we still know little about how it interfaces with the gene regulatory networks underpinning wing formation in insects.

In this work, we performed comparative transcriptomic analysis on SW-destined and LW-destined BPH nymphs, and identified a zinc-finger homeodomain transcription factor Zfh1 that acts as a wing-morph switch in planthoppers. We find that knockdown of *Zfh1* directs SW-destined nymphs to develop into LW morphs by affecting the transcription of *FoxO*. In addition, we find that Zfh1 also acts independently of both the IIS and *Ultrabithorax* pathways in regulating wing-morph development. These findings deepen our understanding of the genetic bases underpinning insect wing polymorphism.

## Results

### Knockdown of *Zfh1* results in LW morph development

To screen potential wing-morph regulators, we conducted RNA sequencing (RNA-seq) on thoracic nota (mesonotum and metanotum) dissected from 0–72 h fifth-instar nymphs of wild-type SW-BPH (*Wt*[SW], SW ratio > 90%) and LW-BPH (*Wt*[LW], LW ratio > 80%) strains, representing SW-destined and LW-destined BPHs, respectively. The fifth-instar stage was chosen because it is the instar during which SW and LW developmental fates are determined[30]. We identified 123 putative transcription factors (TFs) including *FoxO* and *Zfh1* (Fig. 1b, Nl.chr09.837 in Supplementary Data 1) that were significantly differentially expressed in *Wt*[LW] versus *Wt*[SW] nymphs. Next, we performed RNA interference (RNAi)-mediated gene silencing by microinjection of *Wt*[SW] nymphs with double-stranded RNA (dsRNA) targeting each TF gene except for *FoxO* which has been examined previously[22,26]. Microinjection of dsRNA targeting *Zfh1* (ds*Zfh1*, Fig. 1b) reduced the transcriptional level of *Zfh1* by ~40% compared to dsRNA targeting green fluorescent protein (ds*Gfp*, Fig. 1c). Both female (Fig. 1d, e) and male (Supplementary Fig. 1b, c) nymphs treated with ds*Zfh1* exhibited a strong and comparable bias towards LW morphs (ds*Zfh1*[LW]), relative

to control individuals injected with ds*Gfp* (ds*Gfp*[SW]). In contrast, no effect on wing morph determination was detected following dsRNA-mediated knockdown of any of the other 121 TF genes (Supplementary Data 2). Hypomorphic knockdown phenotypes were detectable in a small fraction of dsZfh1-treated adults with intermediate-size wings (ds*Zfh1*[IMW], Fig. 1d, e, Supplementary Fig. 1b, c). Notably, ds*Zfh1* had no detrimental effect on nymph development and survival (Supplementary Fig. 2). Additionally, ds*Zfh1*[LW] BPHs exhibited hind tibia lengths, a common metric for body size in BPH, comparable to *Wt*[LW] BPHs (Fig. 1f, Supplementary Fig. 1d), suggesting that ds*Zfh1* had no obvious effect on body size. However, the forewing size of ds*Zfh1*[LW] BPHs was slightly but significantly smaller than that of *Wt*[LW] BPHs although the vein patterning was within the normal range of variation observed in *Wt*[LW] BPHs (Fig. 1f, g, Supplementary Fig. 1d, e). In addition to wings, ds*Zfh1*[LW] BPHs resembled *Wt*[LW] BPHs with regards to indirect flight muscle (IFM) development. Specially, immunohistochemistry (IHC) staining showed that IFM was abundant only in the thorax of ds*Zfh1*[LW] and *Wt*[LW] BPHs, but not in *Wt*[SW] BPHs (Fig. 2a). Further, examination by transmission electron microscopy (TEM) showed that ds*Zfh1*[LW] and *Wt*[LW] but not *Wt*[SW] BPHs contained well-organized sarcomeres with visible Z and M discs (Fig. 2b). This observation indicates that Zfh1 inhibits IFM differentiation in BPHs, in line with the inhibitory role of Zfh1 in muscle differentiation described in *Drosophila* and vertebrates[31–33].

To confirm the ds*Zfh1* phenotype, we conducted *Zfh1* knockdown using a second non-overlapping dsRNA targeting *Zfh1* (ds*Zfh1_2*, Fig. 1b). Microinjection with ds*Zfh1_2* reduced the transcriptional level of *Zfh1* by ~50% compared to ds*Gfp* treatment (Fig. 1c). As with ds*Zfh1*, the majority of ds*Zfh1_2*-treated *Wt*[SW] nymphs developed into LW adults (Fig. 1e, Supplementary Fig. 1c). Additionally, quantitative real-time PCR (qRT-PCR) analysis showed that *Zfh1* expression exhibited a distinct spatiotemporal pattern, with low level of *Zfh1* detected in the

**Fig. 3 | Knockdown of *L. striatellus* *Zfh* homolog (*LsZfh1*) leads to long-winged (LW) morphs.** Third-instar *L. striatellus* nymphs were microinjected with ds*LsZfh1* or ds*Gfp*. **a** Morphology of ds*Gfp*[SW], ds*LsZfh1*[LW], and ds*LsZfh1*[IMW] females. ds*Gfp*[SW], ds*Gfp*-treated *L. striatellus* planthoppers with short wings. ds*LsZfh1*[LW] and ds*LsZfh1*[IMW], ds*LsZfh1*-treated *L. striatellus* planthoppers with LW and intermediate-size wings (IMW), respectively. **b** Number of *L. striatellus*

planthoppers with different wing morphs upon dsRNA treatments. The LW ratio is compared between two groups using Pearson $\chi^2$ test (****$P = 6.6841\mathrm{E}{-13}$, $\chi^2 = 51.635$ and $df = 1$ for females; ****$P = 0.000001$, and $\chi^2 = 23.413$ and $df = 1$ for males). The experiment was repeated three times independently with similar results. Source data are provided as a Source Data file.

thorax and thoracic nota of fifth-instar $Wt^{LW}$ relative to $Wt^{SW}$ nymphs (Supplementary Fig. 3), indicating that the expression of *Zfh1* is inversely correlated with LW development.

To investigate whether Zfh1 was functionally conserved in the planthopper family Delphacidae, we performed RNAi-mediated knockdown of the *Zfh1* homolog in the planthopper *Laodelphax striatellus* (*LsZfh1*). Microinjection with dsRNA targeting *LsZfh1* (ds*LsZfh1*) significantly decreased the transcriptional level of *LsZfh1* (Supplementary Fig. 4a), and significantly increased the LW ratio (ds*LsZfh1*[LW], Fig. 3a, b) relative to ds*Gfp* treatment, consistent with the phenotype observed in BPHs. Notably, hypomorphic knockdown of *LsZfh1* also led to adults with intermediate-size wings (ds*LsZfh1*[IMW]). Together, these findings indicate that Zfh1 acts as a molecular switch that regulates alternative wing morphs in planthoppers.

## Functional specificity of Zfh1 among BPH homeobox genes

BPH Zfh1 contains a homeodomain in addition to nine Cys2His2 type zinc-finger motifs (Fig. 1b), and as such belongs to the zfh family of homeobox genes across metazoan phyla[34]. As does *Drosophila*, BPH possesses two zfh family members, *Zfh1* and *Zfh2*, which differ in the number and position of homeodomains and zinc-finger motifs. Studies in *Drosophila* indicated that both Zfh1 and Zfh2 are involved in neurogenesis and cell differentiation[35–37]. In addition, *Drosophila* Zfh1 functions in the positioning of mesoderm derivatives including somatic musculature[38], whereas Zfh2 is involved in the proximal-distal patterning of *Drosophila* wing discs[39,40]. Phylogenetic analyses show that both *Zfh1* and *Zfh2* are conserved across animals (Supplementary Fig. 5a): invertebrate *Zfh1* homologs cluster with two vertebrates *Zfh1* orthologues (Zeb1 and Zeb2)[38], whereas *Zfh2* and other zfh family members cluster separately (Supplementary Fig. 5a).

To test whether BPH *Zfh2* is functionally redundant to *Zfh1* with respect to the regulation of wing dimorphism, we conducted *Zfh2*-specific knockdowns by microinjection of fourth-instar $Wt^{SW}$ nymphs with dsRNA targeting *Zfh2* (ds*Zfh2*). Notably, *Zfh2* knockdown caused ~50% mortality of nymphs before adult eclosion, while surviving adults exhibited curved wings (Supplementary Fig. 5b). However, we found that surviving ds*Zfh2*-treated BPHs produced a high proportion of SW morphs, similar to $Wt^{SW}$ BPHs (Supplementary Fig. 5c), indicating that Zfh2 is not involved in BPH wing dimorphism.

To assess whether any additional homeobox genes might be involved in LW development, we conducted RNAi targeting the remaining 87 potential BPH homeobox genes in $Wt^{SW}$ nymphs. We found that knockdown of any of these 87 genes did not change wing-morph ratios (Supplementary Data 3). Thus, our results suggest that *Zfh1* may be the sole BPH homeobox gene that functions specifically in the regulation of wing dimorphism.

## Zfh1 regulates *FoxO* transcriptional activity

To further detail the role of *Zfh1* in the development of wing dimorphism, we collected the mesonotum and metanotum of ds*Zfh1*- and ds*Gfp*-treated fifth-instar nymphs for RNA-seq (Fig. 4a). Interestingly, *FoxO* was one of the 478 genes downregulated following ds*Zfh1* injection (Fig. 4b, Supplementary Data 4). This is notable because ds*Zfh1* phenocopied the effects of *FoxO* knockdown (ds*FoxO*) on LW development[22]. qRT-PCR analysis confirmed that ds*Zfh1* significantly reduced *FoxO* mRNA level (Fig. 4c).

To further assess whether *FoxO* is indeed a target gene transcriptionally regulated by Zfh1, we examined FoxO translation following *Zfh1* knockdown, using a fusion protein consisting of FoxO and Human influenza hemagglutinin (HA) as a reporter. To do so, we microinjected eggs with a mix of Cas9, single-guide RNA (sgRNA), and single-strand DNA (ssDNA) donors to construct a knock-in BPH strain (*FoxO::HA*) using clustered regularly interspaced palindromic repeats/CRISPR-associated (CRISPR/Cas9)-mediated homology-directed repair. The resulting *FoxO::HA* mutant then allowed us to detect FoxO protein levels using commercially available anti-HA antibodies. We found that 13.2% ($n = 1023$) injected egg successfully molted into nymphs (G0), among which 11 out of 44 examined nymphs (25%, Supplementary Table 1) contained an HA-tag at the C terminus of *FoxO*. To examine germline transmission, G0 adults were mated with $Wt^{SW}$ adults to produce G1 offspring. We found that 26.7% G1 individuals (32 out of 120, Supplementary Table 1) contained *FoxO* with an HA insertion (Fig. 4d). Crossing G1 siblings then allowed us to obtain a *FoxO::HA* homozygous mutant. To verify the *FoxO::HA* mutant, whole bodies of *FoxO::HA* mutants and $Wt^{SW}$ adults were homogenized for western blot assay using an HA antibody. We found that the FoxO-HA fusion protein was specifically detected in *FoxO::HA* mutants rather than in $Wt^{SW}$ BPHs (Fig. 4e). To further confirm the *FoxO::HA* mutant, genomic DNA of *FoxO::HA* mutants was extracted and then used for Sanger sequencing with primers (FoxO-KI-F/FoxO-KI-R, Fig. 4d) located outside of homology arms. Sanger sequencing showed that the HA-tag was correctly inserted before the stop codon (TAG) of *FoxO* (Fig. 4f). Next, we performed *Zfh1* knockdown in *FoxO::HA* mutants, followed by western blot assay using an HA antibody. In line with the qRT-PCR data, ds*Zfh1* significantly reduced the FoxO::HA protein expression compared to ds*Gfp* (Fig. 4g).

Given that *FoxO* is transcriptionally regulated by Zfh1, we asked whether Zfh1 protein could bind to the *FoxO* promoter, similar to the vertebrate Zfh1 homolog which represses gene expression by binding to the consensus sequence (CACCT and CACCTG)[41]. We identified three high-affinity Zfh1 binding sites in the first 2.5 kb fragment of the *FoxO* promoter (Pro*FoxO*, Fig. 4h). Using an immunoprecipitation (IP) protein-expressing Zfh1-his in human embryonic kidney 293 cells (HEK293T), we found that the *FoxO* promoter sequence was

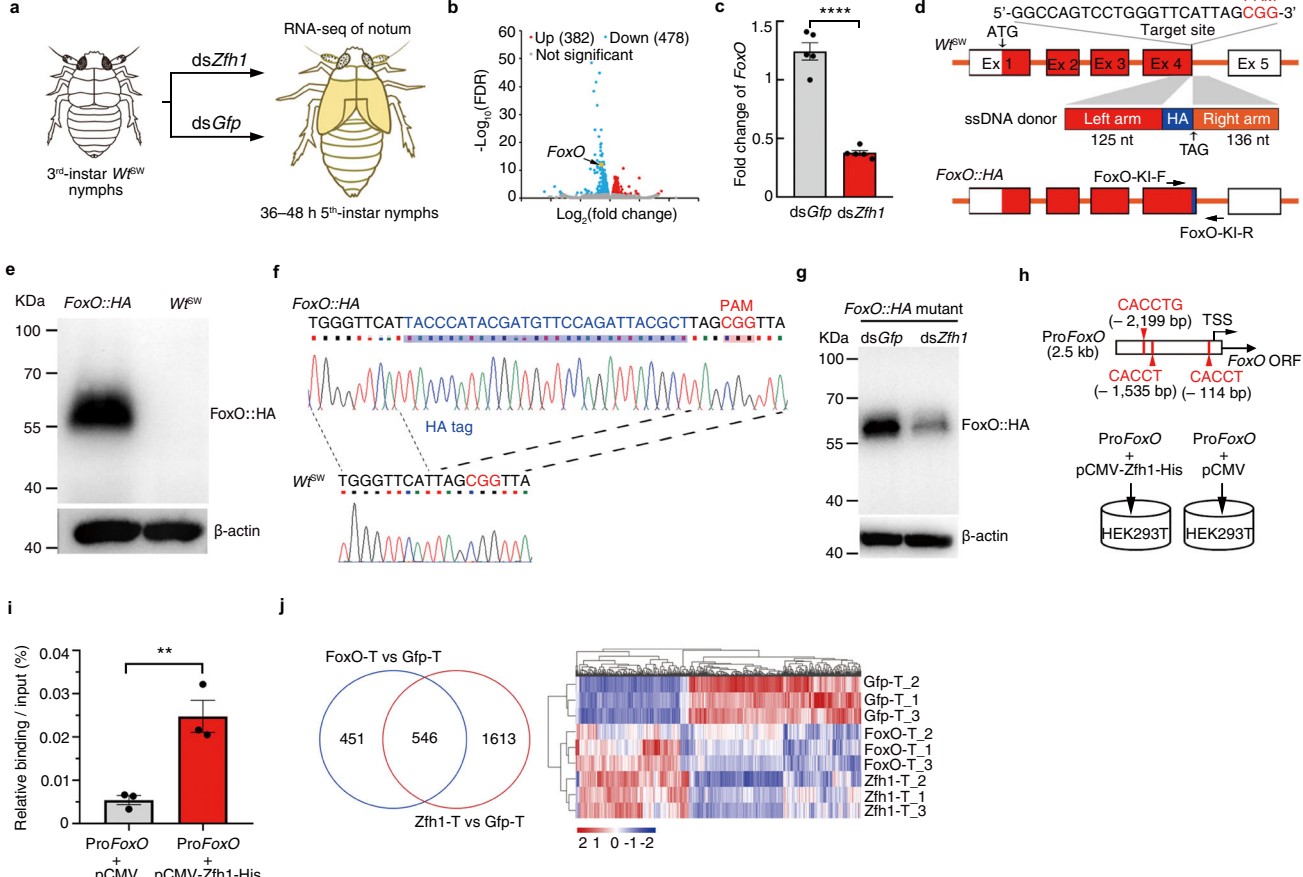

**Fig. 4 | Zfh1 regulates the transcription of *FoxO*. a** Scheme depicting mesonotum and metanotum (yellow) used for RNA-seq (illustrated by Miss Yi Wan). **b** Differentially expressed genes in nota of ds*Zfh1*- and ds*Gfp*-treated fifth-instar nymphs. *FoxO* is indicated by an arrow. **c** qRT-PCR analysis of *FoxO* expression in nota of ds*Zfh1*- and ds*Gfp*-treated fifth-instar nymphs. The experiment was repeated five times with similar results (dots). **d** Scheme of construction of *FoxO::HA* mutants using CRISPR/Cas9-mediated homology-directed repair. The target site for Cas9 and protospacer-adjacent motif (PAM) are indicated. An ssDNA donor contains a left homology arm, a HA-tag (HA), and a right homology arm. Ex 1–5, exons 1–5 of *FoxO*. FoxO-KI-F and FoxO-KI-R, primers used for *FoxO::HA* verification. **e** Confirmation of *FoxO::HA* mutants by western blot using an HA antibody. **f** Confirmation of *FoxO::HA* mutants by Sanger sequencing with primers of FoxO-KI-F and FoxO-KI-R. **g** ds*Zfh1* decreases FoxO::HA protein level. *FoxO::HA* mutants were microinjected with ds*Zfh1* or ds*Gfp*, followed by western blot using an HA antibody. β-Actin is used to show equal protein loading in (**e**)

and (**g**). The experiments in (**e**) and (**g**) were repeated three times independently with similar results. **h** Scheme of immunoprecipitation assay. Red vertical lines and letters, putative Zfh1 consensus binding sites in the *Foxo* promoter (Pro*FoxO*). TSS, the transcription start site of *FoxO*. The HEK293T cells are transfected with the Pro*FoxO* fragment mixed with either pCMV-Zfh1-His or pCMV plasmid. **i** Relative binding of the *Foxo* promoter to Zfh1 by immunoprecipitation assays. The Pro*FoxO* fragment was immunoprecipitated with antibodies against His tag. The experiments were repeated three times with similar results (dots). Two-tailed unpaired *t*-test was used for the statistical analysis (**c**, ****$P = 3.48132E{-}06$) and (**i**, **$P = 0.0075$). Data are presented as mean ± SEM in (**c**) and (**i**). **j** Venn diagram and heat map of thoracic genes commonly regulated by *Zfh1* and *FoxO*. Thoraxes treated with ds*Zfh1*, ds*FoxO*, and ds*Gfp* are denoted as Zfh1-T, FoxO-T, and Gfp-T, respectively. Expression levels are indicated by FPKM (fragments per kilobase of transcript per million mapped reads). Source data are provided as a Source Data file.

significantly enriched by anti-His antibodies when Pro*FoxO* was co-transfected with pCMV-Zfh1-His (Fig. 4i), confirming Zfh1 binding to the *FoxO* promoter.

To further ascertain whether *FoxO* is positioned downstream of *Zfh1*, thoraxes dissected from ds*FoxO*-, ds*Zfh1*-, and ds*Gfp*-treated fifth-instar nymphs were subjected to RNA-seq. The comparative transcriptomic analysis revealed that 55% of genes (546 out of 997) regulated by ds*FoxO* were also regulated by ds*Zfh1* (Fig. 4j, Supplementary Data 5). Among these, 96% (522 out of 546) showed comparable changes in expression (Fig. 4j). Gene ontology (GO) analysis showed important similarities as well as differences among the genes whose expression was significantly altered following ds*FoxO* and/or ds*Zfh1*: genes up-regulated by both ds*FoxO* and ds*Zfh1* were significantly enriched for 9 GO terms associated with cellular processes (Supplementary Fig. 6a), whereas genes downregulated by both ds*FoxO* and ds*Zfh1* were significantly enriched for 10 GO terms associated with chitin metabolic processes (Supplementary Fig. 6b). In contrast, the top GO terms

specific to genes whose expression was altered following ds*FoxO* were 'mitochondrial ribosome', 'organellar ribosome' and 'mitochondrial translation', whereas the top-ranked GO terms specific to ds*Zfh1* regulation were 'structural constituent of cuticle', 'extracellular region' and 'muscle system process' (Supplementary Fig. 6c, d). Taken together, these findings suggest that Zfh1 operates as an upstream regulator for wing dimorphism by affecting *FoxO* transcriptional activity.

**Zfh1 regulates wing morphs in parallel to the IIS pathway**

Inactivation of genes encoding positively acting components (*InR1* or *Akt*) of the IIS pathway can lead to SW morphs by dephosphorylation of FoxO; however, inactivation of *InR2*, a negative regulator of *InR1*, enhances the phosphorylation of FoxO, leading to LW morphs[22]. Because FoxO is an effector downstream of the IIS pathway and because *FoxO* is transcriptionally regulated by Zfh1, we asked whether Zfh1 regulates the expression of *FoxO* by relaying IIS activity. If this is the case, the inactivation of *Zfh1* would abolish any effects of the IIS

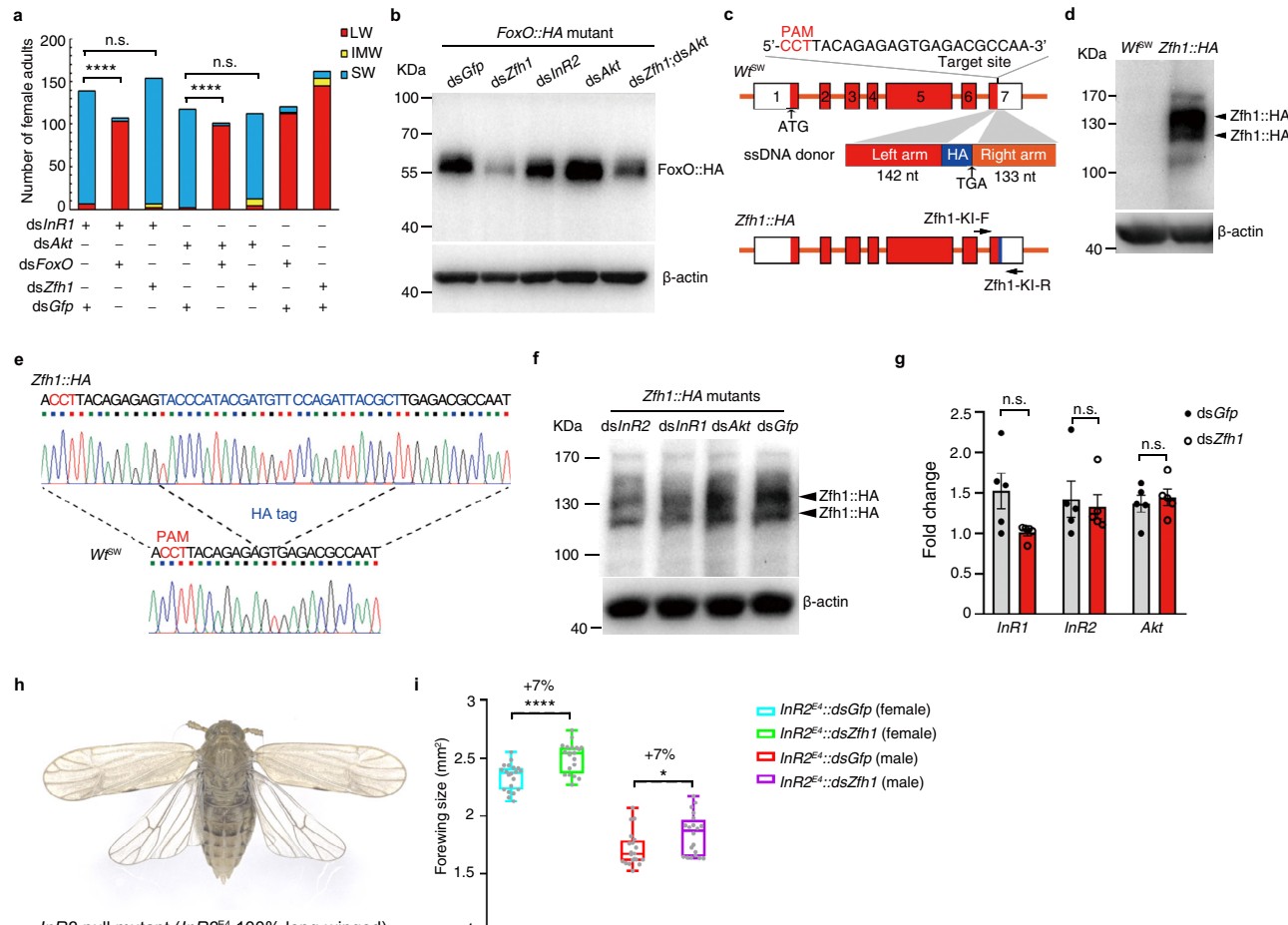

**Fig. 5 | Zfh1 regulates wing dimorphism independent of the IIS pathway.**
**a** Number of females with different wing morphs upon double-gene knockdown. The LW ratio is compared between two groups using Pearson $\chi^2$ test (ds*InR1* + ds*FoxO* vs ds*InR1* + ds*Gfp*: **** $P$ = 5.0866E-47, $\chi^2$ = 207.131, df = 1; ds*Akt* + ds*FoxO* vs ds*Akt* + ds*Gfp*: **** $P$ = 4.763E-45, $\chi^2$ = 198.359, df = 1; n.s. non-significant). **b** The FoxO::HA protein level in *FoxO::HA* mutants following dsRNA treatments. Treatment with ds*Zfh1* only lowers the level of FoxO proteins. **c** Construction of *Zfh1::HA* mutants using CRISPR/Cas9-mediated homology-directed repair. The target site for Cas9, protospacer-adjacent motif (PAM), and exons 1–5 are indicated. An ssDNA donor contains a left homology arm, a HA-tag (HA), and a right homology arm. The primers of Zfh1-KI-F and Zfh-KI-R are designed for Sanger sequencing. **d** Confirmation of *Zfh1::HA* mutants by western blot. Zfh1::HA fusion protein was detected by an HA antibody. **e** Confirmation of *Zfh1::HA* mutants by Sanger sequencing. PCR products amplified from genomic DNA with Zfh1-KI-F and Zfh-KI-R were exposed to Sanger sequencing. The HA

sequence is in blue. **f** The Zfh1::HA protein level in the context of perturbation of the IIS pathway in *Zfh1::HA* mutants. β-actin is used to show equal protein loading in (**b**), (**d**), and (**f**), and t**he** experiments were repeated three times independently with similar results. **g** Relative expression of components of the IIS pathway in ds*Zfh1*-treated BPHs. The experiments were repeated five times with similar results (dots or circles). Data are presented as mean ± SEM. Two corresponding columns are compared using two-tailed unpaired *t*-test (n.s. non-significant). **h** Morphology of an *InR2*-null mutant (*InR2*[E4])[43]. **i** ds*Zfh1* increases the forewing size of *InR2*[E4] mutants. *InR2*[E4] nymphs were microinjected with ds*Gfp* (*InR2*[E4]::ds*Gfp*) or ds*Zfh1* (*InR2*[E4]::ds*Gfp*), and then emerged adults were collected for wing size measurement. The whiskers of box plot represent the quantile percentile, from bottom to top are minima, 25%, median, 75%, and maxima, respectively. Two corresponding boxes are compared using two-tailed unpaired *t*-test (*$P$ = 0.0222 for male and ****$P$ = 5.70008E−05 for female). Source data are provided as a Source Data file.

pathway on wing dimorphism in the same manner as done by *FoxO*[22]. Thus, we performed a double-gene knockdown using ds*FoxO* and ds*Zfh1* in combination with a dsRNA targeting *InR1* (ds*InR1*) or *Akt* (ds*Akt*). Although ds*Zfh1* combined with ds*Gfp* led to LW morphs, *Wt*[SW] nymphs developed into SW adults when ds*Zfh1* was combined with either ds*InR1* or ds*Akt* (Fig. 5a, Supplementary Fig. 7). As a control, *Wt*[SW] nymphs developed into LW adults when ds*FoxO* was combined with ds*InR1* or ds*Akt* (Fig. 5a, Supplementary Fig. 7). These data indicate that Zfh1 is not a TF downstream of the IIS pathway.

Next, we sought to determine whether Zfh1 is positioned upstream of the IIS pathway. If correct, inactivation of *Akt* would abolish the regulatory role of Zfh1 on *FoxO* protein expression. To test these predictions, we microinjected *FoxO::HA* nymphs with ds*Zfh1*, ds*Akt*, a dsRNA targeting *InR2* (ds*InR2*), a dsRNA mixture targeting *Zfh1* and *Akt* (ds*Zfh1*;ds*Akt*), or ds*Gfp*, followed by the dissection of the nota of fifth-instar nymphs for western blot assay. Compared to ds*Gfp*,

double-gene knockdown using ds*Zfh1*;ds*Akt* significantly reduced FoxO::HA fusion protein levels, as did ds*Zfh1* alone (Fig. 5b), indicating that *Zfh1* is also not a factor upstream of the IIS pathway. Notably, both ds*InR2* and ds*Akt* resulted in comparable levels of FoxO::HA protein relative to ds*Gfp* (Fig. 5b). This supports previous reports that the IIS pathway may regulate FoxO protein function via post-translational modification[42].

To exclude any possible crosstalk between the IIS pathway and Zfh1, we generated a *Zfh1::HA* knock-in mutant using CRISPR/Cas9-mediated homology-directed repair, which would allow us to detect Zfh1 protein with anti-HA antibodies since Zfh1 was fused an HA-tag with at the C terminus (Fig. 5c). The CRISP/Cas9-medicated mutagenesis rate and germline transmission for *Zfh1* reached 11.6% ($n$ = 64) and 18.9% ($n$ = 90), respectively. The *Zfh1::HA* homozygous mutant was verified by western blot using anti-HA antibodies (Fig. 5d) and by Sanger sequencing (Fig. 5e) with primers (Zfh1-KI-F/Zfh1-KI-R, Fig. 5c) located outside

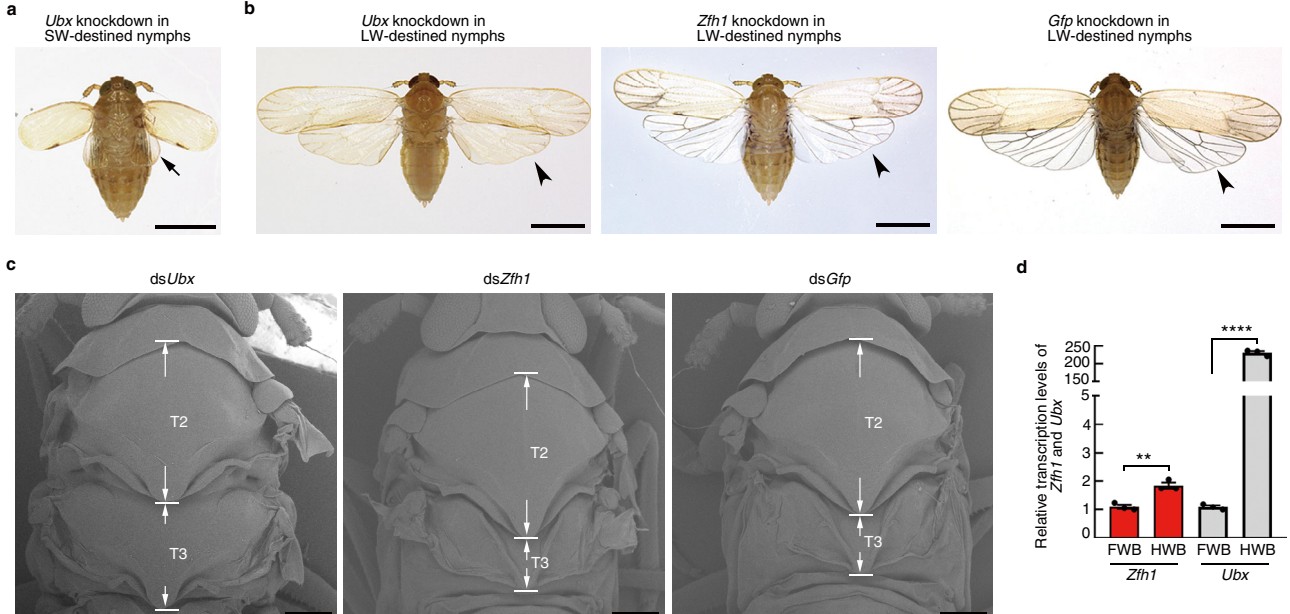

**Fig. 6 | Phenotypic comparison of ds*Zfh1* and ds*Ubx* BPHs. a** Morphology of short-winged (SW) BPHs microinjected with ds*Ubx*. Fourth-instar SW-destined (*Wt*^SW) nymphs were used for microinjection with ds*Ubx*. Small wing-like appendages (hindwings, arrow) protrude from the third thoracic (T3) segment of ds*Ubx*-treated BPHs. Scale bars, 750 μm. **b** Morphology of long-winged (LW) BPHs microinjected with ds*Ubx*, ds*Zfh1*, and ds*Gfp*. LW-destined (*Wt*^LW) nymphs were used for microinjection with ds*Ubx*, ds*Zfh1*, and ds*Gfp*. ds*Ubx*-treated BPHs contain heavily-pigmented hindwings compared to transparently membranous hindwings of ds*Zfh1*- and ds*Gfp*-treated BPHs. Arrowheads, hindwings. Scale bars, 750 μm. **c** Scanning electron microscope observation of the second

thoracic (T2) and T3 nota. Fourth-instar *Wt*^LW nymphs were microinjected with ds*Ubx*, ds*Zfh1* or ds*Gfp*, and emerged adults were directly used for scanning electron microscope. Scale bars, 200 μm. Twenty samples in **a**–**c** were repeated independently with similar results. **d** Examination of the expression of *Ubx* and *Zfh1* in forewing buds (FWB) and hindwing buds (HWB) of fifth-instar nymphs by qRT-PCR. The experiments were repeated three times with similar results (dots). Data are presented as mean ± SEM. Two-tailed unpaired *t*-test was used for the statistical analysis (**$P = 0.004267$ for *Zfh1* and ****$P = 8.11903E-07$ for *Ubx*). Source data are provided as a Source Data file.

of the homology arms. The *Zfh1::HA* mutant allowed us to detect Zfh1 protein in the context of perturbation of the IIS pathway using anti-HA antibodies. We found that knockdown of components of the IIS pathway (*InR1*, *InR2*, and *Akt*) did not affect Zfh1 protein level (Fig. 5f). Meanwhile, knockdown of *Zfh1* did not affect the transcriptional level of *InR1*, *InR2*, and *Akt* (Fig. 5g). Thus, these findings indicate that Zfh1 regulates wing dimorphism independent of the IIS pathway in BPHs.

### Zfh1 functions synergistically with the IIS pathway
Given that Zfh1 regulates LW development in parallel to the IIS pathway, we asked whether Zfh1 and the IIS pathway may interact synergistically in the regulation of LW development. For this purpose, we performed *Zfh1* knockdown in the context of an *InR2*-null mutant (*InR2*^E4)[43], which by itself results in 100% LW morphs (Fig. 5h) due to accumulated phosphorylated FoxO in the cytoplasm via activating the IIS pathway. We found that ds*Zfh1*-treated *InR2*^E4 mutants (*InR2*^E4::ds*Zfh1*) developed forewings that were 7% larger than the *InR2*^E4 mutants treated with ds*Gfp* (*InR2*^E4::ds*Gfp*, Fig. 5i), indicating a moderate synergistic effect of Zfh1 and the IIS pathway on LW development.

### Zfh1 is functionally distinct from *Ultrabithorax*
The presence of a homeodomain in BPH Zfh1 combined with the *Zfh1* knockdown phenotype are reminiscent of the 'four-winged *Drosophila*' phenotype derived from *Ultrabithorax* (*Ubx*) mutants[44]. This phenotype results from the homeotic transformation of the third thoracic segment (T3) to the identity of the second thoracic segment (T2), and the concomitant transformation of halters into hindwings in *Drosophila*[44], or membranous hindwings into elytra in the flour beetle *Tribolium castaneum*[45].

To exclude the possibility that ds*Zfh1* produced LW morphs via homeotic transformation, we compared and contrasted the

morphological characteristics of wings and thoracic segments between ds*Zfh1* and *Ubx*-RNAi (ds*Ubx*) adults, focusing on three aspects: (i) ds*Ubx* adults that were derived from RNAi in SW-destined nymphs developed only small wing-like appendages on T3 and SW forewings at T2 (Fig. 6a), in strikingly contrast to the fully developed forewings and hindwings of ds*Zfh1* BPHs (Fig. 1d); (ii) ds*Zfh1* adults that were derived from RNAi in LW-destined nymphs developed membranous and fully transparent hindwings similar to the control BPHs (ds*Gfp*, Fig. 6b), whereas ds*Ubx* adults that were derived from RNAi in LW-destined nymphs developed heavily-pigmented hindwings (Fig. 6b), resembling forewings; and (*iii*) ds*Zfh1* adults developed T2 and T3 nota identical to those of ds*Gfp*-treated BPHs (Fig. 6c), whereas *Ubx* knockdown led to an expansion of the T3 notum (Fig. 6c), thereby causing it to resemble the characteristics of the T2 notum. In addition, *Zfh1* and *Ubx* differed strikingly in terms of tissue-specific expression. qRT-PCR assay showed that hindwing buds had a slightly but significantly higher level of *Zfh1* relative to forewing buds (Fig. 6d). In contrast, *Ubx* expression was strongly biased toward hindwing buds relative to forewing duds (Fig. 6d). Together, these observations indicate that the SW-to-LW transition derived from *Zfh1* knockdown is most likely not due to homeotic transformation.

### A molecular model of wing dimorphism in planthoppers
Our data revise and significantly extend our understanding of how distinct wing morphs may result from modifications of the same gene regulatory networks (Fig. 7). Under normal conditions, BPHs maintain a basic level of Zfh1, which in turn maintains *FoxO* transcription at low levels. As a result, low activity of the Zfh1-FoxO signaling cascade prevents FoxO from inhibiting wing development, thus leading to LW morphs. However, in response to certain environmental inputs, BPHs raise *Zfh1* transcription, thereby raising FoxO levels, which in turn suppresses LW development and therefore leads to SW morphs. In

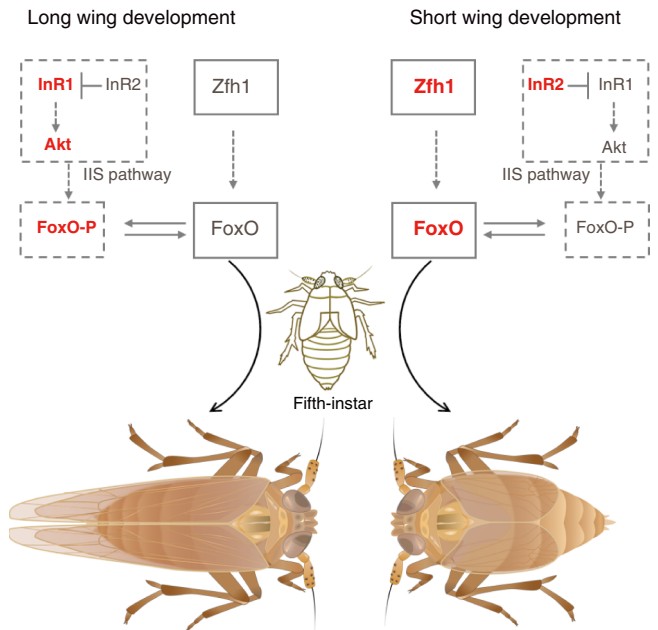

**Fig. 7 | Model of the molecular regulation of wing dimorphism in planthoppers.** The balance of alternative wing morphs is regulated (i) transcriptionally by the Zfh1-FoxO cascade or (ii) post-translationally by the IIS-FoxO cascade. Components with increased expression levels are shown in bold and red. Components with decreased expression levels are shown in gray. FoxO-P, phosphorylated FoxO. The BPH images were illustrated by Miss Xin-Qiu Wang.

addition to this Zfh1-FoxO signaling cascade, wing dimorphism is regulated by a second, and independent, signaling axis at the post-translational level: activation of the IIS pathway phosphorylates FoxO and thus exports FoxO from the nucleus to the cytoplasm, leading to LW morphs. In contrast, inactivation of the IIS pathway dephosphorylates FoxO and thus accumulates FoxO in the nucleus, leading to SW morphs.

## Discussion

We identified the zinc-finger homeodomain transcription factor Zfh1 as a wing-morph molecular switch that determines long or short wings by regulating the transcriptional level of a second transcription factor FoxO in BPHs. Previous studies indicated that *Zfh1* is an evolutionarily conserved gene critical for counteracting the myogenic differentiation program[31,34,46]. In *Drosophila*, embryos with a *Zfh1* loss-of-function mutation show alterations in somatic muscles[38]. In the same way, a vertebrate homolog of Zfh1 (ZEB) acts as a negative regulator of muscle differentiation[31]. Here, we found that knockdown of BPH *Zfh1* promoted IFM development, consistent with myogenesis suppression by *Zfh1* in *Drosophila* and vertebrates. However, in addition to these common phenotypes reported previously, this study demonstrates for the first time that knockdown of *Zfh1* directs SW-destined nymphs to develop into LW in wing dimorphic BPHs, and likely planthoppers broadly. Given that the development of LW in BPHs mainly relies on cell proliferation[23,43], this phenotype indicates that Zfh1 might act as an inhibitor of wing cell proliferation. However, whether suppression of wing cell via Zfh1 is species-dependent or conserved among insects remains unknown. Additionally, a recent study showed that the maintenance of *Drosophila* muscle progenitors involves a switch between *Zfh1*-long and *Zfh1*-short RNA isoforms[33]. The *Zfh1*-long isoform is subjected to *miR-8* microRNA-mediated downregulation that enables muscle progenitors to differentiate in early pupal stages, whereas the *Zfh1*-short isoform cannot be targeted by *miR-8*, thus allowing these cells to persist as muscle progenitors in the

adult stage[33]. In this study, only one *Zfh1* isoform was identified after examining all BPH transcriptomes prepared by our lab. Thus, whether multiple BPH *Zfh1* isoforms, if they do indeed exist, possess distinct roles in the regulation of wing development and muscle homeostasis needs to be clarified in further studies.

Our previous studies showed that FoxO plays a pivotal role in controlling the development of alternative wing morphs in BPHs via relaying the activity of the IIS pathway[22]. Activation of the IIS pathway phosphorylates and deactivates FoxO, leading to LW morphs, while IIS inactivation and FoxO dephosphorylation promote SW morphs. Here, we found that knockdown of Zfh1 stimulates LW development by decreasing *FoxO* transcriptional activity. These findings highlight that the Zfh1 and IIS cascades may converge on FoxO to regulate wing dimorphism in BPHs. This assumption is supported by a synergistic effect of both pathways on LW development. However, exactly how BPHs balance the relative activities of the Zfh1-FoxO and IIS-FoxO cascades to regulate wing developmental plasticity remains to be determined. More generally, our results suggest that this regulatory system may provide a multi-tiered regulatory machinery able to facilitate evolutionary changes in threshold responses separating wing morphs and increase the ways in which population and species may diverge genetically in the precise regulation of wing polyphenism.

## Methods

### Insects

Laboratory populations of *N. lugens* were established from colonies collected in Hangzhou, China, in 2008. The $Wt^{SW}$ colony was purified by inbreeding for more than 13 generations[47]. The $Wt^{LW}$ strain was provided by Dr. Hong-Xia Hua (Huazhong Agricultural University, China). The planthopper *L. striatellus* was collected in Hangzhou, China, 2018. All insects were reared at 26 ± 0.5 °C under a photoperiod of 16:8 h (L:D) at a relative humidity of 50 ± 5% on rice seedlings (rice variety: Xiushui 11).

### Sample preparation for RNA-seq

To screen for potential wing-morph regulators, mesonotum and metanotum were dissected from 0–72 h fifth-instar $Wt^{SW}$ ($n = 20$) and $Wt^{LW}$ nymphs ($n = 20$), denoted as 5th-SW and 5th-LW respectively (BioProject ID: PRJNA805393). To identify downstream effectors targeted by Zfh1, third-instar $Wt^{SW}$ nymphs were microinjected with 50 ng ds*Zfh1* or ds*Gfp*. Subsequently, mesonotum and metanotum were dissected from 36–48 h fifth-instar nymphs with three replicates ($n = 30$ for each replicate), denoted as Zfh1-nota and Gfp-nota (BioProject ID: PRJNA805395). To investigate common genes regulated by *Zfh1* and *FoxO*, third-instar $Wt^{SW}$ nymphs were microinjected with 50 ng ds*Zfh1*, ds*FoxO*, or ds*Gfp*. Thoraxes were dissected from 48–54 h fifth-instar nymphs with three replicates ($n = 30$ for each replicate), denoted as Zfh1-T, FoxO-T and Gfp-T (BioProject ID: PRJNA805400). Samples were homogenized for total RNA isolation using RNAiso Plus (Cat#9109, Takara) according to the manufacturer's protocol.

### cDNA library preparation and Illumina sequencing

A total of 1 μg RNA per sample was used to construct a sequencing library using a NEBNext Ultra RNA library prep kit for Illumina (Cat#E7770, NEB) according to the manufacturer's recommendations, and index codes were added to each sequence. Library fragments of 250–300 bp in length were preferentially purified using an AMPure XP system (Beckman Coulter). Clustering of the index-coded samples was performed on a cBot Cluster generation system using a TruSeq PE cluster kit v3-cBot-HS (Cat#PE-401-3001, Illumina) according to the manufacturer's instructions. The cDNA libraries were sequenced on an Illumina Novaseq 6000 platform and 150 bp paired-end reads were generated.

## Read mapping and differentially expressed genes (DEGs)

After Illumina sequencing, clean reads were generated by removing adapters, poly-N, and low-quality reads from the raw data using fastp algorithm (v0.12.4)[48]. Clean reads were mapped against the *N. lugens* genome v4.0[47] using hisat2 (v2.1.0)[49], and transcript abundance was quantified using StringTie (v1.3.5)[50]. Fragments Per Kilobase of transcript per Million mapped reads (FPKM) was used to quantify the expression level of each transcript. The read count information of each transcript was extracted directly from the files generated by StringTie using a python script of prepDE.py (http://ccb.jhu.edu/software/stringtie/dl/prepDE.py). Differential expression analysis of 5th-SW and 5th-LW samples was conducted using the edgeR package (v3.38.4)[51], and genes with a false discovery rate (FDR) < 0.01 were considered to be significant. Differential expression analysis of Zfh1-nota versus Gfp-nota, Zfh1-T versus Gfp-T, and FoxO-T versus Gfp-T was conducted using the DESeq2 package (v1.36.0)[52] and gene expression change was considered to be significant under the FDR < 0.05. To generate heat map, FPKM of each replicate of Zfh1-T, FoxO-T and Gfp-T were z-score transformed and clustered using the online OmicShare tool (https://www.omicshare.com/tools/Home/Soft/heatmap). GO enrichment analysis of DEGs was performed using the online OmicShare tool (https://www.omicshare.com/tools/home/report/goenrich.html), and the enriched GO terms with FDR ≤ 0.05 were considered as significant.

## RNAi and RNAi efficiency

DsRNA synthesis and injection were performed as previously described[22]. Briefly, primers were synthesized with T7 RNA polymerase promoter at both ends (Supplementary Data 6), and then dsRNAs were synthesized using a T7 high yield RNA transcription kit (Cat#TR101-02, Vazyme) according to the manufacturer's instructions. Microinjection was performed using a FemtoJet microinjection system (Eppendorf). Each third-instar or fourth-instar nymph was microinjected with approximately 50 ng or 100 ng dsRNAs, respectively. Two days after microinjection, insects ($n = 5$ for each of three replicates) were collected for RNAi efficiency examination by qRT-PCR. The ribosomal protein S11 gene (*rps11*)[53] and ribosomal protein L5 gene (*rpl5*)[54] were used as the internal reference gene in BPH and *L. striatellus* planthopper, respectively.

## qRT-PCR assay

Total RNAs were isolated from BPHs using RNAiso Plus (Cat#9109, Takara) according to the manufacturer's protocol. The first-strand cDNA was synthesized from total RNAs (900 ng) using HiScript QRT super mix (Cat#R123-01, Vazyme). The qRT-PCR was conducted on a CFX96 real-time PCR detection system (Bio-Rad) with the following conditions: denaturation for 3 min at 95 °C, followed by 40 cycles at 95 °C for 10 s, and then 60 °C for 30 s. The *rps11* and *rpl5* were used as the internal reference gene in BPH and *L. striatellus* planthopper, respectively. The $2^{-\Delta\Delta Ct}$ method (Ct represents the cycle threshold) was used to measure relative expression levels. Three biological replicates were used for statistical comparison between samples.

## Nymphal duration and survival rate

Third-instar $Wt^{SW}$ nymphs were microinjected with 50 ng ds*Zfh1* or ds*Gfp*. The developmental times of fourth- and fifth-instar stages and survival rate were monitored every 12 h. To determining the nymphal duration, 13 females and 19 males were used for ds*Zfh1*, and 17 females and 11 males were used for ds*Gfp*. To conduct the survival assay, 50 nymphs were used for ds*Zfh1* and ds*Gfp*.

## Spatiotemporal expression of *Zfh1*

To investigate the temporal expression of *Zfh1* in $Wt^{LW}$ and $Wt^{SW}$ BPHs, total RNAs were isolated from the thorax of first-instar ($n = 100$), second-instar ($n = 50$), third-instar ($n = 50$), fourth-instar ($n = 30$), fifth-

instar nymphs ($n = 15$), and female adults ($n = 15$). To investigate the spatial expression of *Zfh1*, fifth-instar $Wt^{LW}$ and $Wt^{SW}$ nypmphs ($n = 50$) were collected for tissue dissection, and then total RNA was isolated from head, fat body, the abdominal cuticle, six legs, the whole digestive tract (gut), and nota (mesonotum and metanotum), respectively. To investigate the expression pattern of *Zfh1* across the fifth-instar stage, mesonotum and metanotum were dissected from 24 h-, 48 h-, and 72 h-fifth-instar $Wt^{LW}$ and $Wt^{SW}$ nymphs ($n = 50$), and then used for total RNA isolation. Three independent biological replicates were used for RNA isolation, and the first-strand cDNA was synthesized for *Zfh1* quantification. The relative expression level of *Zfh1* was normalized to that of the *rps11* gene.

## IHC staining

IHC staining was performed as previously described[55]. Briefly, third-instar nymphs were microinjected with 50 ng ds*Zfh1*. Samples were fixed using 4% paraformaldehyde in PBS overnight at 4 °C, following by blocking with Tissue-Tek O.C.T. Compound (Cat#4583, Sakura Finetek) at −80 °C. Samples were longitudinally cut into ~30 μm sections using a Lecia CM1900 cryotome (Leica Microsystems) at −20 °C, and then transferred to Superfrost⁺ slides (Cat#12-550-15, Thermo Fisher Scientific). The cytoskeleton and nucleus were stained using 100 nM rhodamine-labeled phalloidin (Cat# 40734ES75, Yeasen) and 100 nM DAPI (Cat#D9542, Sigma Aldrich), respectively. Fluorescence images were acquired using a Zeiss LSM 800 confocal microscopy (Carl Zeiss MicroImaging).

## TEM assay

Third-instar nymphs were microinjected with ds*Zfh1*. Thoraxes were dissected from 24 h ds*Zfh1*$^{LW}$ females for transmission electron microscopy as previously described[55]. Briefly, samples were fixed in 2.5% glutaraldehyde overnight at 4 °C, followed by post-fixation in 1% osmium tetroxide for 1.5 h. The samples were then dehydrated in a standard ethanol/acetone series, infiltrated and embedded in Spurr medium, and then superthin sections (~70 nm) were cut. The sections were stained with 5% uranyl acetate followed by Reynolds' lead citrate solution and observed under a JEM-1230 transmission electron microscope (JEOL) at the Analysis Center of Agrobiology and Environmental Sciences of Zhejiang University.

## Phylogenetic analysis of Zfh1 and Zfh2

BlastP (v2.2.31) was used to search Zfh1 and Zfh2 homologs using the *N. lugens* Zfh1 and Zfh2 as query sequences. Multiple amino acid sequence alignment was performed using the MUSCLE algorithm with MEGA-X (v10.1.8) software. The phylogenetic tree was constructed by MEGA-X using maximum-likelihood methods under the JTT + G model with 1000 bootstrap replicates. Phylogenetic tree was annotated and edited using the iTOL tools (https://itol.embl.de/) and Adobe Photoshop CC (v19.1.9) program.

## In vitro synthesis of sgRNA and Cas9 mRNA

sgRNA was designed as previously reported[56]. Briefly, sgRNA was designed by manually searching genomic sequence around the region of the *FoxO* and *Zfh1* stop codon for the sequences corresponding to 5′-N$_{17-20}$NGG-3′, where NGG is the protospacer-adjacent motif (PAM) of SpCas9 and N is any nucleotide. For in vitro transcription of sgRNA, a DNA fragment was amplified by PCR from pMD19-T sgRNA scaffold vector with a forward primer containing a T7 promoter and a reverse primer containing a partial sgRNA sequence. The PCR products were used as a template for in vitro transcription using a T7 high yield RNA transcription kit (Cat#TR101-02, Vazyme) at 37 °C overnight according to the manufacturer's instruction.

In vitro synthesis of Cas9 mRNA was prepared as previously reported[57]. Cas9 mRNA was transcribed from plasmid pSP6-2sNLS-SpCas9 vector using the mMESSAGE mMACHINE SP6 transcription kit

(Cat#AM1340, Thermo Scientific) and Poly(A) tailing kit (Cat#AM1350, Thermo Scientific).

## Construction of ssDNA donors

The ssDNA was produced via lambda exonuclease digestion from PCR amplicons according to a previous description[58] with minor modifications. Briefly, a dsDNA fragment containing the whole ssDNA donor cassette for HA-tag knock-in was de novo synthesized and then subcloned into a cloning vector. For *FoxO::HA* knock-in mutants, the ssDNA donor cassette consisted of a 125 nt left homology arm (LHA) upstream of the *FoxO* stop codon, a 136 nt right homology arm (RHA) downstream of the *FoxO* stop codon, and a HA-tag (5′-TACCCA-TACGATGTTCCAGATTACGCT-3′) in the middle region. For *Zfh1::HA* knock-in mutant, the ssDNA donor cassette consisted of a 142 nt LHA upstream of the *Zfh1* stop codon, a 133 nt RHA downstream of the *Zfh1* stop codon, and a HA-tag in the middle region. Then, the dsDNA fragments were used as a template for PCR amplification with a forward primer and a 5′ phosphorylated reverse primer (Supplementary Data 6). PCR products were treated with lambda exonuclease (Cat#EN6501, Thermo Scientific) to remove the 5′-phosphorylated DNA. The ssDNAs were purified by phenol-chloroform extraction, followed by isopropanol precipitation, and then dissolved nuclease-free water.

## Embryo microinjection and CRISPR/Cas9-mediated homology-directed repair

Microinjection of BPH embryos was performed as previously described[57] with minor modifications. Briefly, pre-blastoderm eggs were collected within 1 h after oviposition, and microinjected with a solution mixture (-0.5 nl) containing 500 ng/μl Cas9 mRNA, 300 ng/μl sgRNA and 300 ng/μl ssDNA using a FemtoJet microinjection system (Eppendorf). To minimize mechanical damage, the posterior end of the eggs was pierced with glass needles. After microinjection, eggs were transferred to a sterile petri dish covered with filter papers that were moistened with an antibiotic solution containing tebuconazole (20 ng/ml) and kanamycin (50 ng/ml). The eggs were incubated at 30 °C and allowed to hatch into nymphs (G0).

## Crossing and genotyping

DNA typing of *FoxO:: HA* and *Zfh1::HA* heterozygous or homozygous mutants was determined as described previously[57]. Newly enclosed G0 adults were mated with *Wt*$^{SW}$ adults to produce G1 progeny. After mating, one individual G0 adult was homogenized in a 1.5-ml sterile Eppendorf tube containing 100 μl of DNA extraction buffer (10 mM Tris-HCl pH 8.2, 1 mM EDTA, 25 mM NaCl, and 0.2 mg/ml proteinase K) as reported previously[59]. The body lysate was incubated at 37 °C for 30 min, followed by incubation at 95 °C for 2 min to inactivate proteinase K. Then, the precipitation was removed by centrifuging at 15,000 × *g* for 2 min at room temperature, and the supernatant was kept for PCR amplification.

To determine the genotyping of G1 BPHs, wings were dissected from a single adult and then incubated in 100 μl wing gDNA extraction buffer (10 mM Tris-HCl pH 8.0, 10 mM EDTA, 100 mM NaCl, 39 mM DTT, 2% SDS, and 0.02 mg/ml proteinase K) overnight at 37°C. gDNA was precipitated in 100 μl isopropanol, followed by washing with 500 μl ethanol (80%). The gDNA precipitation was dissolved in ddH$_2$O, and then used for PCR amplification.

To confirm the correct insertion of the HA sequence in *FoxO::HA* mutants, PCR was conducted with primer pairs of FoxO-KI-F/HA-R or FoxO-KI-F/FoxO-KI-R (Supplementary Data 6). To confirm the correct insertion of the HA sequence in *Zfh1::HA* mutants, PCR was conducted with primer pairs of Zfh1-KI-F/HA-R or Zfh1-KI-F/Zfh1-KI-R (Supplementary Data 6). The FoxO-KI-F/FoxO-KI-R and Zfh1-KI-F/Zfh1-KI-R primer pairs were located outside of homology arms and the HA-R primer was located inside the HA encoding sequence. The PCR

products were either directly used for Sanger sequencing or subcloned into cloning vectors, and then single clones were picked for Sanger sequencing. Those G1 individuals with the correct HA-tag insertion were collected and sibling-crossed to obtain homozygous *FoxO::HA* and *Zfh1::HA* strains.

## Western blot analysis

To detect FoxO::HA or Zfh1:HA fusion protein, whole body of fourth-instar nymphs (*n* = 40) or thorax (mesonotum and metanotum) dissected from 48 h fifth-instar nymphs (*n* = 35) was pooled for homogenization. Samples were homogenized in RIPA lysis and extraction buffer (Cat#89900, Thermo Scientific), and then denatured in SDS-PAGE loading buffer. Equal amounts of protein were loaded for each lane on SDS-PAGE gel and then transferred to a polyvinylidene difluoride membrane (Cat#IPVH00010, Millipore). The membranes were blocked in TBST (TBS with 0.1% Tween-20) containing 5% non-fat dried milk powder overnight at 4 °C. After washing, the membranes were incubated with anti-HA mAb (Cat# M180-3, MBL, 1: 2000) for 1.5 h at 35 °C, followed by incubation with horseradish peroxidase (HRP) conjugated goat anti-mouse antibody (Cat#SA00001-1, Proteintech, 1: 5000) for 1 h at 35 °C. The antibody against β-actin (Cat#M1210-2, Huabio, 1:5000) was used as the loading control. Secondary antibody binding was detected using the chemiluminescence HRP substrate detection method (Cat#34577, Thermo Scientific) and imaged with the Molecular Imager ChemiDoc XRS system (Bio-Rad).

## Determination of the full-length cDNA of *Zfh1*

Total RNA was isolated from the whole body of fifth-instar *Wt*$^{SW}$ BPHs (*n* = 10) using RNAiso plus (Cat#9109, Takara). First-strand cDNA was synthesized using a HiScript-TS 5′/3′ RACE kit (Cat#RA101-01, Vazyme) according to the manufacturer's instruction. To obtain the 5′ end sequence of *Zfh1*, PCR was conducted with primers of Universal-primer and Zfh1-5RACE-GSP1, followed by a nested PCR with primers of nested-primer and Zfh1-5RACE-GSP2 (Supplementary Data 6). To obtain the 3′ end sequence of *Zfh1*, PCR was conducted with primer of Universal-primer and Zfh1-3RACE-GSP1, followed by a nested PCR with primers of Nested-primer and Zfh1-3RACE-GSP2 (Supplementary Data 6). All amplicons were subcloned into cloning vectors and confirmed by Sanger sequencing (Tsingke Biological Technology). The cDNA sequence of *Zfh1* was deposited in GenBank with accession number OM283826.

## Double-gene RNAi

Third-instar *Wt*$^{SW}$ nymphs were microinjected with 50 ng ds*Zfh1* or ds*FoxO*, followed by microinjection with 50 ng ds*InR1*, ds*Akt*, or ds*Gfp* at the fourth-instar stage. In parallel, fourth-instar *Wt*$^{SW}$ nymphs were microinjected with dsRNA mixtures of ds*Akt* and ds*Gfp* (50 ng each) or ds*InR1* and ds*Gfp* (50 ng each). All dsRNA-treated nymphs were raised to adults, and BPHs with different wing morphs (LW, SW, and IMW) were collected for statistical analysis.

## IP assay

To express *Zfh1* in HEK293T cell, the recombinant fragment contained *Zfh1* ORF and 6*His tag were cloned downstream of a CMV promoter, designed pCMV-Zfh1-His. A genomic DNA fragment (Pro*FoxO*) located 4,606-bp upstream of the start codon (ATG) of *Zfh1* was amplified by PCR with primers of ProFoxO-F and ProFoxO-R (Supplementary Data 6), which contains a 607-bp 5′UTR of Zfh1 and putative *FoxO* promotor region. IP assays were performed using IP assay kit (Cat#P2078, Beyotime) as per the manufacturer's protocol. Briefly, HEK293T cells were transfected with Pro*FoxO* in combination with pCMV-Zfh1-his or pCMV vector alone. After 36 h, the transfected cells were cross linked using 1% formaldehyde before cell lysis. Sonicated lysates were incubated overnight at 4°C with 6*His tag monoclonal antibody (Cat#66005-1-Ig, 1: 650 (4 μg), Proteintech). The Pro*FoxO*

fragments were quantified by qRT-PCR with ChamQ SYBR color qPCR master mix (Cat# Q411-02, Vazyme) using primers of qChIP-F and qChIP-R (Supplementary Data 6).

### Relative expression of *InR1*, *InR2*, *Akt*, and *FoxO* in the context of *Zfh1* knockdown

Third-instar nymphs were microinjected with 50 ng ds*Zfh1* or ds*Gfp*. Mesonotum and metanotum were dissected from 36–48 h fifth-instar nymphs ($n = 25$) and then used for total RNA isolation. The first-strand cDNA was synthesized with random primers and then used as templates for qRT-PCR. The relative expression level of *InR1*, *InR2*, *Akt*, and *FoxO* was normalized to that of the *rps11* gene. Five independent biological replicates with three technical replicates were conducted for each experiment.

### Relative expression of *Zfh1* and *Ubx* in wing buds

Wing buds on the second and third thoracic segments were dissected from 48 h fifth-instar *Wt*^SW nymphs ($n = 150$), and then used for RNA isolation. The first-strand cDNA was synthesized with random primers and then used as templates for qRT-PCR with primers specific to *Zfh1* and *Ubx* (Supplementary Data 6). The relative expression level of *Zfh1* was normalized to that of the *rps11* gene. Three independent biological replicates with three technical replicates were conducted for each experiment.

### Scanning electron microscope (SEM)

Fourth-instar *Wt*^LW nymphs were microinjected with ds*Ubx*, ds*Zfh1*, or ds*Gfp*, and allowed to molting into adults. After removing wings, female adults were placed on a stub and dried in an ion sputter (Hitachi) under vacuum. After gold sputtering, the samples were observed under SEM (TM-1000, Hitachi).

### Image acquisition

Images of insects were taken using a DVM6 digital microscope (Leica Microsystems) with LAS X software. Images of wings and tibia were captured with a DFC320 digital camera attached to a Leica S8AP0 stereomicroscope using the LAS (v. 3.8) digital imaging system. Digital images of forewings ($n = 20$) and hind tibias ($n = 20$) were collected for measurement of forewing size and hind tibia length using ImageJ (v. 1.47). Plotting and statistical analysis were performed with GraphPad Prism (v8.0.1).

### Reporting summary

Further information on research design is available in the Nature Research Reporting Summary linked to this article.

## Data availability

All data are available in the manuscript or the supplementary materials. The DNA sequencing data generated in this study have been deposited in the GenBank database under the following accession codes: OM283826 (the *Zfh1* gene), XP_039284941.1 (the *Zfh2* gene), OM676634 (the *LsZfh1* gene). The RNA sequencing data generated in this study have been deposited under the following accession coded: PRJNA805393, PRJNA805395, and PRJNA805400. Source data are provided with this paper.

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

## Acknowledgements

H.J.X. was supported by the National Natural Science Foundation of China (31972261, 32272519, and 31772158), Key Projects of Natural Science Foundation of Zhejiang Province (LZ21C140002), and Development and Demonstration of Comprehensive Technology for Prevention and Control of Major Diseases and Pests (2021YFD1401100). J.L.Z. was supported by the National Natural Science Foundation of China (32102186). We thank Prof. Hongxia Hua in Huazhong Agricultural University for providing the *Wt*^LW strain.

## Author contributions

J.L.Z. and H.J.X. conceived the study and developed the initial study design. J.L.Z., S.J.C., and X.Y.L. performed the experiments. J.L.Z. and H.J.X. analyzed the experiments. A.P.M. and H.J.X. wrote the manuscript. All authors contributed intellectually to the experiments as well as editing and approval of the final version of the manuscript.

## Competing interests

The authors declare no competing interests.
