## [Peer Review File · Nature Communications]

The transcription factor Zfh1 acts as a wing-morph switch in planthoppersREVIEWER COMMENTS

Reviewer #1 (Remarks to the Author):

Key results

The manuscript by Zhang et al. titled "The transcription factor Zfh1 acts as a wing-morph switch in planthoppers" is a genetic characterization of the function of Zfh1 on wing development in an insect species that exhibits phenotypic plasticity for wing development based on environment conditions. Brown rice planthopper (BPH) is an excellent model system to study the genetic regulation of wing morph polyphenism because these insects develop either short wings or long wings in response to environmental cues such as nutrition condition and crowding. The insulin signaling pathway has been shown in many studies by different groups of researchers to be primary signal transduction cascade that turns the environmental cue into an adaptive developmental response. Short wing morphs invest more in ovary and egg production and stay where conditions are favorable (high food quality and low crowding). Long wing morphs in contrast are the migratory form and are induced by low food quality and high density such that the insects fly away to find more optimal conditions. This is an interesting study because through a transcriptomics analysis they have identified this transcription factor, Zfh1 that, when the transcript is knocked down, induces long wing morph formation in the absence of environmental cues.

The brown rice planthopper is the main study organism although a brief mention and one RNAi experiment is included on a different planthopper species which is presumably why the authors make the larger claim in the title of planthoppers more generally. The main findings of this study are (1) that Zfh1 functions to promote short wing morph development in two species of planthopper and (2) that Zfh1 acts independently of the Ubx pathway and the insulin signaling pathway in regulating wing morph development.

Validity & Significance

This is an interesting and important new contribution to the understanding of the genetic regulation of wing polyphenism in insects and the genetics underlying the evolution of adaptive traits in general.

Significance

Data and methodology

These seem straightforward, although not well justified in some places and seem out of order and context in others.

Suggested improvements

The descriptions of the wing morphology in Figure 1 and the subsequent discussion should be clarified. Figure 1a shows 1 short pair of fore-wings on T2 and no second pair of hindwings on T3. Terminology and descriptions should be consistent such that RNAi knockdown results can be compared across all morph types and all gene knockdown types.

Figure 5 is really confusing. "Wing development by default" should be changed to Long Wing development and Short Wing development. No where in the manuscript is the idea that short wings are a derived form and long wings are the default state introduced before this figure. These labels without description confuse the main findings of this study, that "active Zfh1" results in short wing morphs. This language in the caption of "active" versus inactive is confusing and should be made more specific. It's not clear what Figure 5 is trying to say, this should be re-done or left out.

Extended Data Table 2 does not include any sample sizes. The sample sizes are necessary.

Clarity and context

Despite the significance of the scientific contribution of the study, the paper is poorly organized and lacks justification or rationale for the experiments. There was a general lack of description of wing

polyphenism in general and especially wing development. For the non-expert, this makes the significance of the paper very difficult to grasp. The introduction also jumps directly into what FoxO is doing but it is confusing to be introduced to this so early in the manuscript.

Figure captions were not clear or very explanatory. The most important revision should be to use direct language for the effect of Zfh1 on wing development. The abstract and the introduction suffered from the use of this particular phrase "... knockdown of Zfh1 redirects wing development from SW to LW morphs." While this might be one way to explain it, it is used repeatedly and for the general audience, the idea of "redirection" seems wrong. Once you have delved into the experimental design and realize that the authors have used a selected line that mostly results in SW morphs, then this phrase makes sense. But in a general sense, it would also be correct to say that Zfh1 knockdown results in LW morph formation. If that's not true, then it is difficult, or even impossible, to get a clear understanding of the intention of the authors on this. A BPH nymph has the potential to have SW or LW but this idea of "redirection" is really confusing.

Another issue is that there are some interesting experiments that are not well justified. For example, the CRISPR/Cas9 knock-in experiment described in lines 135 on was not well justified. While this is an exciting example of the use of this technology, it was unclear what the experiment was contributing that was not already shown by the qPCR results. Granted it shows that the protein was knocked down as was the transcript, but this was not explained as a reason to perform the experiment.

It is not clear from the Ubx and Zfh1 experiments that the conclusion that the Zfh1 knockdown is not due to a homeotic transition is needed for the main findings of this study. It's interesting, but not well justified as to why it is included.

References

Lack of relevant citations from the Xinda Lin group on insulin signaling in brown rice planthoppers.
Lack of relevant citations on Ubx genetic function.

Reviewer #2 (Remarks to the Author):

This study unveiled an unexpected new player in the regulation of polyphenism in the planthopper. How environment-dependent polyphenism is genetically regulated is one of the attractive topics in developmental biology for a long time. Although ecological and physiological studies on this topic are rich, underlying genetic and evolutionary mechanisms remain elusive. This study deploys cutting-edge technologies such as deep sequencing and genome editing in the planthopper to tackle the question, and successfully found a novel regulator Zfh1 that parallelly regulates FoxO to form the wing dimorphism. I appreciate the authors' effort to perform knockdowns of over a hundred genes to characterize this regulator. The in-depth study is effectively designed to posit the role of this new player independent of the canonical IIS signaling pathway. This finding implicates another unknown axis that parallelly regulates the environment-dependent dimorphism, which is insightful for understanding evolution and development of polyphenism. I thus suggest publication of this work after addressing the minor points below.

Line 58

Which stage of tissues from the 5th instar nymphs were sampled for this RNA-seq?

Line 95

A brief description of known functions of Zfh1 in other insects would be helpful to understand the functional novelty here.

Line 117, Fig. 2b

Although the authors' conclusion here is collectively convincing, it is unclear why the authors conclude that the T3 identity is unchanged in Zfh1-RNAi insects from the provided information. I suggest adding annotations on Fig. 2b to indicate characteristic anatomical features on T2 and T3. Electron micrography might be helpful here.

Lines 135 and 180

It would be informative if the authors provide a summary table that shows the efficiency of CRISPR/Cas-mediated gene knock-in in the planthopper because it is still challenging in many species.

Fig. 3h

This result certainly supports that Zfh1 regulates FoxO, but large non-overlapping regions suggest independent functions might also contribute to form the dimorphism. To clarify this point, it would be helpful to show what kind of transcripts were influenced by each RNAi treatment—what is overlapped and what is not between zfh1 and FoxO RNAi. Gene set analysis or a similar analysis is useful.

Extended Data Table 2

How many individuals were injected with each dsRNA in this experiment?

Extended Data Table 2, Extended Data Fig. 7

It would be informative to show more detail of zfh2, such as structure and pictures of RNAi individuals, to understand the evolutionary distance between zfh1 and zfh2.

Extended Data Table 3

The authors should disclose all primer sequences.

Entire manuscript

Please check grammar and words again.

e.g.

Line 32 particularly → particular

Line 187 inwhich → in which

Reviewer #3 (Remarks to the Author):

This manuscript characterizes the role of zfh1 in planthopper wing morph development. Wing polyphenism is a fascinating case of developmental plasticity in general, and it has been a classical topic in insect biology. This study uses a wide range of genetic and molecular methods to address its central claim, and I believe it makes a sound case. I have few general comments I hope the authors and editors will consider, and several minor points.

Major comments

- Right now this manuscript is very focused. For a general readership, I suggest the authors take a few sentences at the beginning to make the case for the importance of this work beyond insect biology. Right now, the first paragraph cites the right literature, but does a poor job of connecting this study to its broader evolutionary and developmental context.
- It's worth noting that the thoroughness of the experiments here is quite impressive. The number of RNAi experiments alone is substantial: nearly 200 treatments. The in-knock HA-tags using CRISPR really helps provide evidence for their model.
- I do wonder at the absence of phenotypes for some dsRNAs though. Ubx RNAi gives a classic phenotype, but RNAi targeting the Hox genes Dfd, Scr, Antp and abdB does not produce any homeotic phenotype? How should we interpret other negative results? Could there be other regulators of wing morphs?
- I had some concerns about the LW and SW "destined" genotypes. It is stated at one point (lines 60-

61) that these genotypes produce an 80-90% skew in morphs. But when nymphs are chosen for RNAseq or other experiments, can they be certain there aren't future long-winged BPH among the WtSW group? Some commentary on the diluting influence of this fact would be prudent.

- It would be useful to clarify the role of sex in wing morph determination. From other BPH polyphenism papers, I understand there to be a small sex-effect, but it's not clear whether this is persistent for each genotype. I note that some experiments in this study focus only on females or males. It's not clear to me why that's the case. Some justification is necessary.
- The term "hierarchical regulator" used in the first paragraph and elsewhere in this manuscript seems like the unnecessary invention of a novel term. I would prefer to see *zfh1* referred to as "upstream" of FoxO and "parallel" to IIS, etc.
- Given the complexity of the dataset presented here, I do not wish to suggest more experiments. However, it does occur to me that their methods predicts that FoxO protein concentrations should differ between the nucleus and cytoplasm of each morph.
- I find the model presented in Fig 5 satisfying, and a good reflection of the data in this manuscript and other recent ones in their field. However, I would appreciate some speculation on upstream factors influencing *zfh1*. How might it be regulated? Is it likely to be environmentally regulated at all? If this is an evolutionarily conserved mechanisms, as they suggest, how or why might this point of regulation have arisen?

Specific comments

With reference to line numbers

- 12-13: "...escaping predators, and locating mates." Presumably, their arthropod ancestors could do those things too. Just cut this phrase.
- 15: "leading to" the logic here is questionable. Does flight loss precede brachyptery? It seems that cannot be stated with certainty. In any case, it's besides the point being made in this manuscript.
- 17: "...for studying the evolutionary significance of dispersal." I'm not sure statement is true actually. Roff and Harrison's theoretical work was foundational, but dispersal ecology has, as far I can tell, not used polymorphic species as models. In any case, that's not the focus of this manuscript.
- 19-20: Cut "hierarchical regulator"
- 34: persuasive, not "persuading"
- 34-36: There is some grammatical issue here that makes the meaning of this sentence unclear.
- 37: aphids, plural
- 41: Publication in 2015 no longer seems "recent"!
- 70: rather than
- 105: grammar issue here
- 124-5: "*zfh1* transcripts were equally distributed in T2 and T3 wing buds" -- The figure shows 1.7 fold differences actually. Admittedly this is much less than for *Ubx*. However, I would caution the authors against comparing the magnitude of differences here. The relative realtime PCR (using the delta-delta-Ct method) is comparative. The 100% on their y-axis could represent essentially zero transcripts, making such comparisons spurious.
- 306: Define IMW
- Extended Data Fig 4: This is useful data, but perhaps present images more zoomed out to provide more anatomical context.
- 596: What software was used for read filtering?

Point-by-point response to reviewers

Comments from Reviewer #1

Reviewer #1 (Remarks to the Author):

Key results

The manuscript by Zhang et al. titled “The transcription factor Zfh1 acts as a wing-morph switch in planthoppers” is a genetic characterization of the function of Zfh1 on wing development in an insect species that exhibits phenotypic plasticity for wing development based on environment conditions. Brown rice planthopper (BPH) is an excellent model system to study the genetic regulation of wing morph polyphenism because these insects develop either short wings or long wings in response to environmental cues such as nutrition condition and crowding. The insulin signaling pathway has been shown in many studies by different groups of researchers to be primary signal transduction cascade that turns the environmental cue into an adaptive developmental response. Short wing morphs invest more in ovary and egg production and stay where conditions are favorable (high food quality and low crowding). Long wing morphs in contrast are the migratory form and are induced by low food quality and high density such that the insects fly away to find more optimal conditions. This is an interesting study because through a transcriptomics analysis they have identified this transcription factor, Zfh1 that, when the transcript is knocked down, induces long wing morph formation in the absence of environmental cues.

The brown rice planthopper is the main study organism although a brief mention and one RNAi experiment is included on a different planthopper species which is presumably why the authors make the larger claim in the title of planthoppers more generally. The main findings of this study are (1) that Zfh1 functions to promote short wing morph development in two species of planthopper and (2) that Zfh1 acts independently of the Ubx pathway and the insulin signaling pathway in regulating wing morph development.

Validity & Significance

This is an interesting and important new contribution to the understanding of the genetic regulation of wing polyphenism in insects and the genetics underlying the evolution of adaptive traits in general.

Significance

Comment 1: Data and methodology

These seem straightforward, although not well justified in some places and seem out of order and context in others.

Response to comment 1: To better justify experiments and improve clarity and accessibility of the manuscript we made the following modifications:

- (1) We moved the paragraph of “**The SW-to-LW transition in dsZfh1 BPHs is not due to homeotic phenotype**” to the second paragraph from the bottom, and changed the subtitle into “**Zfh1 is functionally distinct from Ultrabithorax**”.
- (2) We separated the section on homeobox-gene-RNAi from the paragraph of “**Zfh1 is functionally distinct from Ultrabithorax**”, and gave it a separate, independent subtitle (“**Functional specificity of Zfh1 among BPH homeobox genes**”). This new paragraph now follows the first paragraph of “**Knockdown of Zfh1 results in LW morph development in SW-destined nymphs**”.
- (3) We separated the section on Zfh1 knockdown in the context of an *InR2*-null mutant from the paragraph of “**Zfh1 regulates wing morphs in parallel to the IIS pathway**”, and assigned it an independent subtitle (“**Zfh1 and the IIS pathway function synergistically in the regulation of LW development**”).
- (4) We explained experimental design and the resulting data in more detail
- (5) We provided more experimental data to better clarify how we arrive at our conclusions.
- (6) We adjusted several paragraphs in “Methods” in accordance with the main text.

Comment 2: Suggested improvements

The descriptions of the wing morphology in Figure 1 and the subsequent discussion

should be clarified. Figure 1a shows 1 short pair of fore-wings on T2 and no second pair of hindwings on T3. Terminology and descriptions should be consistent such that RNAi knockdown results can be compared across all morph types and all gene knockdown types.

Response to comment 2: In the revised manuscript, we have provided a magnified figure to show hindwings of SW adults (Fig. 1a). This figure shows vestigial forewings and rudimentary hindwing buds in SW adults, in strikingly contrast to fully-developed forewings and hindwings in LW adults (lines 64-69 in the Xu_manuscript_R1_clean copy).

To keep terminology and descriptions consistent, we have replaced T2- and T3-wings with forewings and hindwings across the manuscript.

Comment 3: *Figure 5 is really confusing. “Wing development by default” should be changed to Long Wing development and Short Wing development. No where in the manuscript is the idea that short wings are a derived form and long wings are the default state introduced before this figure. These labels without description confuse the main findings of this study, that “active Zfh1” results in short wing morphs. This language in the caption of “active” versus inactive is confusing and should be made more specific. It’s not clear what Figure 5 is trying to say, this should be re-done or left out.*

Response to comment 3: Thank you for this suggestion. We have changed “Wing development by default” into “Long wing development” in the revised manuscript (Fig. 7). In the figure caption, we changed “active components...” and “Less or inactive components...” into “Components with increased expression levels...” and “Components with decreased expression levels...”, respectively.

In 2015, we found that FoxO acts as a wing-morph switch in planthoppers (Xu et al. 2015. Nature 519, 464-467), and proposed a model to explain how the insulin signaling pathway regulates wing dimorphism via modulating the phosphorylation level of FoxO. In this work, the purpose of Figure 5 (Figure 7 in the Xu_manuscript_R1_clean copy) is meant to help readers understand how the new

findings report here updated and nuance the molecular regulation of wing dimorphism in planthoppers as depicted by the original model. We would prefer to keep this model within the publication, but if you and the editor feel strongly it must be revised further we would of course be happy to do so.

Comment 4: *Extended Data Table 2 does not include any sample sizes. The sample sizes are necessary.*

Response to comment 4: The “Extended Data Table 2” has been changed to “Supplementary Table 2” in the revised manuscript. We have added sample sizes in “Supplementary Table 2”. In brief, for each microinjection with dsRNAs, 150 fourth-instar *Wt^{SW}* nymphs were used.

Comment 5:

Clarity and context

Despite the significance of the scientific contribution of the study, the paper is poorly organized and lacks justification or rationale for the experiments. There was a general lack of description of wing polyphenism in general and especially wing development. For the non-expert, this makes the significance of the paper very difficult to grasp. The introduction also jumps directly into what FoxO is doing but it is confusing to be introduced to this so early in the manuscript.

Figure captions were not clear or very explanatory. The most important revision should be to use direct language for the effect of Zfh1 on wing development. The abstract and the introduction suffered from the use of this particular phrase ...” knockdown of Zfh1 redirects wing development from SW to LW morphs.” While this might be one way to explain it, it is used repeatedly and for the general audience, the idea of “redirection” seems wrong. Once you have delved into the experimental design and realize that the authors have used a selected line that mostly results in SW morphs, then this phrase makes sense. But in a general sense, it would also be correct to say that Zfh1 knockdown results in LW morph formation. If that’s not true, then it is difficult, or even impossible, to get a clear understanding of the intention of the authors on this.

A BPH nymph has the potential to have SW or LW but this idea of “redirection” is really confusing.

Response to comment 5: Thank you for this insightful suggestion. In response, we divided and re-organized the paragraphs, and explained experimental design and data in more detail. Please see the changes in the manuscript with track changes (Xu_Manuscript_R1_with track changes) and the manuscript with clean copy (Xu_Manuscript_R1_clean copy).

To better help non-specialist readers understand the goals of this manuscript, we now provide more background about BPH biology and wing development in the introduction (lines 62-73 in the Xu_manuscript_R1_clean copy). Furthermore, we have revised figure captions as the reviewer suggested. We believe these changes will help readers to better understand goals, findings, and implications of this manuscript.

Comment 6: *Another issue is that there are some interesting experiments that are not well justified. For example, the CRISPR/Cas9 knock-in experiment described in lines 135 on was not well justified. While this is an exciting example of the use of this technology, it was unclear what the experiment was contributing that was not already shown by the qPCR results. Granted it shows that the protein was knocked down as was the transcript, but this was not explained as a reason to perform the experiment.*

Response to comment 6: Thank you for your suggestion. We agree that in the originally submitted manuscript, we did not explain well why we executed knock-in experiments, in part because of word limits. In the revised manuscript, we now devote more time to explain the motivation behind our knock-in experiments and address the mutagenesis rate in more detail (lines 183-197 and 262-267 in the Xu_manuscript_R1_clean copy). In brief, we believe that at least two independent experimental approaches are required to confidently verify gene regulation. This is why we executed both qPCR and western blot-based approaches to investigate whether FoxO is indeed regulated by Zfh1 or not. Thus, the main reason we created knock-in

mutants was because antibodies against BPH FoxO and Zfh1 were not commercially available, yet *FoxO::HA* and *Zfh1::HA* mutants helped us easily detect FoxO and Zfh1 proteins using HA antibodies.

Comment 7: *It is not clear from the Ubx and Zfh1 experiments that the conclusion that the Zfh1 knockdown is not due to a homeotic transition is needed for the main findings of this study. It's interesting, but not well justified as to why it is included.*

Response to comment 7: We agree that on first sight this may not be obvious. However, three reasons ultimately compelled us to compare the phenotypes derived from *Zfh1* and *Ubx* knockdown and to include the results in this ms: (1) both *Zfh1* and *Ubx* are homeobox genes characterized by the presence of a homeodomain. (2) *Ubx* mutant causes a transformation of halteres into hindwings in *Drosophila*. Similar to the *Ubx* phenotype, *Zfh1* knockdown causes hindwing buds to develop into wings in BPH. (3) A recent report claimed that *Ubx* is a key regulator for switching between LW and SW morphs in BPH (Liu et al., 2020. *National Science Review* 7). However, our data now clearly show that *Ubx* causes homeotic transformations instead of wing polyphenism, and *Zfh1* functions are distinct from those of *Ubx* (Fig. 6b, c). Since our results clearly contradict claims made by others in the literature we do feel they should be included in this ms.

Liu, F., Li, X., Zhao, M. et al. Ultrabithorax is key regulator for the dimorphism of wings, a main cause for the outbreak of planthoppers in rice. *National Science Review* 7: 1181-1189 (2020).

Comment 8:

References

Lack of relevant citations from the Xinda Lin group on insulin signaling in brown rice planthoppers. Lack of relevant citations on Ubx genetic function.

Response to comment 8:

We cited two papers on the Ubx function in *Drosophila* and the flour beetle *Tribolium castaneum* in the revised manuscript (lines 286-292 in the the Xu_manuscript_R1_clean copy).

(Ref. 41) Lewis, E. B. A gene complex controlling segmentation in *Drosophila*. *Nature* 276, 565–570 (1978).

(Ref. 42) Tomoyasu, Y., Wheeler, S. R. & Denell, R. E. Ultrabithorax is required for membranous wing identity in the beetle *Tribolium castaneum*. *Nature* 433, 643-647 (2005).

Dr. Xinda Lin graduated from our institute about 20 years ago. Now he is working in a university not far away from me, and sometimes we discuss our findings. Dr. Xinda Li published some interesting and important findings on BPH wing dimorphism in the past several years although some need to be further clarified. In our revised manuscript, we cited five papers from Xinda Lin's group.

(Ref. 23) Lin, X. et al. Cell cycle progress determines wing morph in the polyphonic insect *Nilaparvata lugens*. *iScience* **23**, 101040 (2020).

This paper showed that wing cells in SW-destined BPHs were largely in the G2/M phase of the cell cycle, whereas those in in LW individual (FoxO RNAi) were largely in G1. This finding is consistent with our data published in 2021 (Xue et al., 2021. PLoS Genetics 17: e1009653.)

(Ref. 24) Lin, X. et al. JNK signaling mediates wing form polymorphism in brown planthoppers (*Nilaparvata lugens*). *Insect Biochem. Mol. Biol.* **73**, 55-61 (2016).

This paper showed that JNK knockdown moderately increased the proportion of SW females (from 87.7% to 98.1%) compared to dsGfp treatment, and the dsJNK effect was FoxO dependent. However, the dsJNK effect was sex-specific as this phenotype was not observed on males. Although he did not discuss the reason of the sex-specific effect, we speculated this might be due to the BPH population he used in his experiment.

(Ref. 25) Lin X., Yao, Y., Wang, B., Lavine, M. D. & Lavine, L. C. FOXO links wing form polyphenism and wound healing the brown planthopper, *Nilaparvata lugens*. *Insect Biochem. Mol. Biol.* **70**, 24-31 (2016).

This paper showed that wounding significantly increased the proportion of SW morphs in both females and males, and this effect was mediated by FoxO. We found that the wounding effect could be observed in some BPH populations, but in a stable BPH population raised in a lab for a long time.

(Ref. 26) Hayes, A. M., Lavine, M. D., Gotoh, H., Lin, X. & Lavine, L. C. Mechanisms

regulating phenotypic plasticity in wing polyphenic insect. *Adv. In Insect Phys.* **56**, 43-72 (2019).

(Ref. 27) Lin, X., Yao, Y., Wang, B., Emlen, D. J. & Lavine, L. C. Ecological trade-offs between migration and reproduction are mediated by the nutrition-sensitive insulin-signaling pathway. *Int. J. Biol. Sci.* **12**, 607-616 (2016).

This paper showed that manipulation of components of the insulin signaling pathway changed BPH wing morphs. This observation supports our data published in 2015, showing that the insulin signaling pathway modulates wing morphs via FoxO (Xu et al., 2015. *Nature* 519, 464-467). However, different phenotypes were obtained between Dr. Lin's group and ours when a component of the IIS pathway was knocked down. Lin's group showed that knockdown of *Akt* led to significant wing-morph ratio changes in females but not in males, in striking contrast to almost 100% SW females and males derived from *Akt* RNAi in our experiment (Xu et al. 2015. *Nature* 519: 464-467)

Here, we want to emphasize some important background information and characteristics of BPH biology that might be easily overlooked: (1) Lab-raised BPH populations tend to develop into SW morphs. The LW ratio varies from generation to generation, and even from different batches taken from the same generation. So, a LW ratio based on small sample size, although statistically significant, should be interpreted with caution; (2) Several LW-destined populations were set up by several labs from China and Japan. To keep the LW ratio at a high level, LW-females and LW-males must be picked out from a population to produce offspring in each generation. So, it's time-consuming to maintain a LW population; (3) A lab-raised SW-BPH population is refractory to environmental cues that hypothetically stimulate LW development. However, SW-BPHs freshly collected from rice fields still keep the capability of responding to environmental cues. This observation is in line with a newly published data from an independent group in Nanjing Agriculture University (Zhang et al., 2022. *Insect Science*. doi: 10.1111/1744-7917.13037); (4) In native BPH populations, males have LW ratio ~10% higher than females. As a consequence, wing-morph ratios need to be assessed in females and males separately.

Comments from Reviewer #2

Reviewer #2 (Remarks to the Author):

*This study unveiled an unexpected new player in the regulation of polyphenism in the planthopper. How environment-dependent polyphenism is genetically regulated is one of the attractive topics in developmental biology for a long time. Although ecological and physiological studies on this topic are rich, underlying genetic and evolutionary mechanisms remain elusive. This study deploys cutting-edge technologies such as deep sequencing and genome editing in the planthopper to tackle the question, and successfully found a novel regulator *Zfh1* that parallelly regulates *FoxO* to form the wing dimorphism. I appreciate the authors' effort to perform knockdowns of over a hundred genes to characterize this regulator. The in-depth study is effectively designed to posit the role of this new player independent of the canonical IIS signaling pathway. This finding implicates another unknown axis that parallelly regulates the environment-dependent dimorphism, which is insightful for understanding evolution and development of polyphenism. I thus suggest publication of this work after addressing the minor points below.*

Comment 1: Line 58

Which stage of tissues from the 5th instar nymphs were sampled for this RNA-seq?

Response to comment 1: To screen potential wing-morph regulators, we conducted RNA sequencing (RNA-seq) on thoracic nota (mesonotum and metanotum) dissected from 0–72 h old fifth-instar nymphs of wild-type SW-BPH (WtSW, SW ratio > 90%) and LW-BPH (WtLW, LW ratio > 80%) strains, representing SW-destined and LW-destined BPHs, respectively (lines 99-102 in the Xu_manuscript_R1_clean copy).

Comment 2: Line 95

A brief description of known functions of *Zfh1* in other insects would be helpful to understand the functional novelty here.

Response to comment 2: In the revised manuscript, a paragraph with the subtitle “*Zfh1*

is functionally distinct from *Ultrabithorax*” is used to introduce the functional difference between *Zfh1* and *Ubx* (lines 286-292 in the Xu_manuscript_R1_clean copy). We cited two papers on the *Ubx* function in *Drosophila* and the flour beetle *Tribolium cataneum* in the revised manuscript.

Comment 3:

Line 117, Fig. 2b

Although the authors’ conclusion here is collectively convincing, it is unclear why the authors conclude that the T3 identity is unchanged in Zfh1-RNAi insects from the provided information. I suggest adding annotations on Fig. 2b to indicate characteristic anatomical features on T2 and T3. Electron micrography might be helpful here.

Response to comment 3: Thank you for your suggestion. We employed scanning electron microscopy to examine the thoracic nota of *dsUbx*-, *dsZfh1*-, and *dsGfp*-treated BPHs. *Zfh1* knockdown had T2 and T3 nota identical to *dsGfp*-treated BPHs (Fig. 6c). In contrast, *Ubx* knockdown led to an expansion of the T3 notum (Fig. 6c), thereby causing it to resemble the characteristics of the T2 notum (lines 303-306 in the Xu_manuscript_R1_clean copy).

Comment 4:

Lines 135 and 180

It would be informative if the authors provide a summary table that shows the efficiency of CRISPR/Cas-mediated gene knock-in in the planthopper because it is still challenging in many species.

Response to comment 4: We have provided a table (Supplementary Table 3) to show the mutagenesis rate (Lines 191-196 and lines 265-267 in the Xu_manuscript_R1_clean copy).

Comment 5:

Fig. 3h

This result certainly supports that Zfh1 regulates FoxO, but large non-overlapping regions suggest independent functions might also contribute to form the dimorphism.

To clarify this point, it would be helpful to show what kind of transcripts were influenced by each RNAi treatment—what is overlapped and what is not between zfh1 and FoxO RNAi. Gene set analysis or a similar analysis is useful.

Response to comment 5: In response we performed Gene ontology (GO) analysis on genes commonly affected by FoxO and Zfh1, or specifically regulated by FoxO or Zfh1 (Supplementary Fig. 6, lines 222-235 in the Xu_manuscript_R1_clean copy). We hope this formation will help readers better understand the diversity of genes regulated by Zfh1 and FoxO.

Comment 6:

Extended Data Table 2

How many individuals were injected with each dsRNA in this experiment?

Response to comment 6: The “Extended Data Table 2” has been changed to “Supplementary Table 2” in the revised manuscript. We have added the sample size in “Supplementary Table 2”. For each microinjection with dsRNAs, 150 fourth-instar Wt^{SW} nymphs were used.

Comment 7:

Extended Data Table 2, Extended Data Fig. 7

It would be informative to show more detail of zfh2, such as structure and pictures of RNAi individuals, to understand the evolutionary distance between zfh1 and zfh2.

Response to comment 7: To test whether BPH *Zfh2* is functionally redundant to *Zfh1* with respect to the regulation of wing dimorphism, we conducted *Zfh2*-specific knockdowns by microinjection of fourth-instar Wt^{SW} nymphs with dsRNA targeting *Zfh2* (*dsZfh2*). Notably, *Zfh2* knockdown caused ~50% mortality of nymphs before adult eclosion, while surviving adults exhibited curved wings (Supplementary Fig. 5b). However, we found that surviving *dsZfh2*-treated BPHs produced a high proportion of SW morphs, similar to Wt^{SW} BPHs (Supplementary Fig. 5c), indicating that *Zfh2* is not involved in BPH wing dimorphism (lines 162-169 in the Xu_manuscript_R1_clean copy).

Comment 8:

Extended Data Table 3

The authors should disclose all primer sequences.

Response to comment 8: We have provided all primers in the Supplementary Data 4.

Comment 9:

Entire manuscript

Please check grammar and words again.

e.g.

Line 32 particularly -> particular

Line 187 inwhich -> in which

Response to comment 9: Thank you for the reminder. We have checked grammar and spelling across the manuscript.

Comments from Reviewer #3

Reviewer #3 (Remarks to the Author):

*This manuscript characterizes the role of *zfh1* in planthopper wing morph development. Wing polyphenism is a fascinating case of developmental plasticity in general, and it has been a classical topic in insect biology. This study uses a wide range of genetic and molecular methods to address its central claim, and I believe it makes a sound case. I have few general comments I hope the authors and editors will consider, and several minor points.*

Comment 1:

Major comments

- Right now this manuscript is very focused. For a general readership, I suggest the authors take a few sentences at the beginning to make the case for the importance of this work beyond insect biology. Right now, the first paragraph cites the right literature, but does a poor job of connecting this study to its broader evolutionary and developmental context.

Response to comment 1: Thank you for your suggestion. To broaden the introduction and improve clarity and accessibility of the manuscript overall, we have made major revisions, from the abstract all the way to figure captions, as well as included additional citations. Please see Xu_manuscript_R1_clean copy and Xu_manuscript_R1_with track changes for details. We hope these revisions address the reviewer's concerns.

Comment 2:

- It's worth noting that the thorough-ness of the experiments here is quite impressive. The number of RNAi experiments alone is substantial: nearly 200 treatments. The in-knock HA-tags using CRISPR really helps provide evidence for their model.

*- I do wonder at the absence of phenotypes for some dsRNAs though. Ubx RNAi gives a classic phenotype, but RNAi targeting the Hox genes *Dfd*, *Scr*, *Antp* and *abdB* does not produce any homeotic phenotype? How should we interpret other negative results? Could there be other regulators of wing morphs?*

Response to comment 2: Thank you for this comment – this is an issue we struggled with quite a bit. We executed both nymphal and parental RNAi targeting Hox genes including *Lab*, *Pb*, *Dfd*, *Scr*, *Antp*, *Abd-A*, and *Abd-B* in addition of *Ubx*. We observed diverse interesting phenotypes in response, however, we did not include these into our manuscript because none of them seemed related to the overall goals of this manuscript. The phenotype is briefly as follows:

- (1) To conduct nymphal RNAi, fourth-instar *Wt^{SW}* nymphs were used for RNAi, and adults were collected for morphological examination: (a) Knockdown of *Lab*, *Dfd*, and *Antp* had no discernable effects on organismal development and wing morphs (Response Letter Fig. 1); (b) Knockdown of *Scr* caused a high mortality, and surviving adults had thin bodies (Response Letter Fig. 1); (c) Knockdown of *Pb* and *Abd-A* resulted in 100% lethality before adult eclosion (Response Letter Fig. 1); (d) Knockdown of *Abd-B* also caused high mortality, and only 5% (n = 270) nymphs survived to adults; and (e) Knockdown of *Abd-B* had no effect on wing morphs, but caused a genitalia-to-leg transformation in both females and males as we reported recently (Chen et al. 2022. *Insect Molecular Biology* 4: 447-456), akin to results generated for holometabolous insects (e.g. Stansbury and Moczek 2012, *Proc Roy Soc* 281: 1471-2954). Since these phenotypes were derived from RNAi in 4th-instar nymphs, more severe defects might be obtained if earlier instar nymphs were used for RNAi.
- (2) To conduct parental RNAi, newly emerged females and males were microinjected with dsRNAs and then mated to deposit eggs. Phenotypes obtained were as follows: (a) Parental RNAi of *Scr* caused an extra thoracic segment and a transformation of piercing-sucking mouthparts into leg-like appendages in offspring nymphs (Response Letter Fig. 2; matching gnathal mouthpart transformations in e.g. beetles; Wasik et al. 2010, *Evolution & Development* 12:353-362); (b) Parental RNAi of *pb* caused a transformation of piercing-sucking mouthparts into leg-like appendages, but did not affect thoracic segment in offspring nymphs (Response Letter Fig. 2); (c) Parental RNAi of *Lab* and *Dfd* caused malformed growth of the segment where compound eyes attached in

offspring nymphs (Response Letter Fig. 2); **(d)** Parental RNAi of *Ubx* caused an additional leg-like appendages on the first abdomen of offspring nymphs (Response Letter Fig. 2). We have published this data in 2020 (Fu et al., 2020. *Gene* 737: 144446); **(e)** Parental RNAi of *Abd-B* caused an extra abdominal segment in offspring nymphs (Response Letter Fig. 3); and **(f)** Parental RNAi of *Abd-A* led to embryos with leg-like appendages on the second-to-eight abdominal segment (Response Letter Fig. 4). We have published this data in 2022 (Chen et al. 2022. *Insect Molecular Biology* 4: 447-456).

(3)

Response Letter Fig. 1. Nymphal RNAi of Hox genes. Knockdowns of *Antp*, *Dfd*, and *Antp* have no discernable on BPH development and wing morphs. Knockdowns of *Pb*, *Scr*, *Abd-A*, and *Abd-B* cause a high lethality before adult eclosion. A fraction of surviving adults possesses short wings as control BPHs (*Gfp* RNAi)

Response Letter Fig. 2. Parent RNAi of *Scr*, *Lab*, *Pb*, *Dfd*, and *Ubx*. Thorax segments (T1-T3) and piercing-sucking mouthpart (arrowhead) in control BPHs (*Gfp* RNAi) are indicated. Four thorax segments (instars) and leg-like appendages (arrows) are derived from *Scr* parental RNAi. Malformed head is derived from *Lab* and *Dfd* parental RNAi. Leg-like appendages (arrows) are derived from *Pb* parental RNAi. Extra leg-like appendages (arrows) on the first abdomen are derived from *Ubx* parental RNAi. L1-L3, legs on the first, second and third thoracic segment.

Response Letter Fig. 3. Parental RNAi of *Abd-B*. an extra abdominal segment is produced upon *Abd-B* parental RNAi. Abdominal segment in *Gfp*-RNAi BPHs (A1-A9)

and in *Abd-B* BPHs (A1-A10) are indicated.

Response Letter Fig. 4. Parental RNAi of *Abd-A*. Extra leg-like appendages on abdominal segments of embryos are indicated by arrowheads, and the corresponding abdominal segments (A2-A8) are labelled. The legs on the thorax are indicated by stars and thorax segments are labelled (T1-T3).

Comment 3:

- I had some concerns about the LW and SW "destined" genotypes. It is stated at one point (lines 60-61) that these genotypes produce an 80-90% skew in morphs. But when nymphs are chosen for RNAseq or other experiments, can they be certain there aren't future long-winged BPH among the WtSW group? Some commentary on the diluting influence of this fact would be prudent.

Response to comment 3: The reviewer raises an important issue. As far as we know, no 100% SW and 100% LW BPH populations have been reported so far. The SW population (SW >90%) and the LW population (LW > 80%) used in our study are the best material we could get.

We agree with the reviewer that some information might be obscured by rare morphs contained within these nearly fully SW and LW destined populations. The only way we can deal with this is to use very large sample sizes compared to many other studies, which we have done in this work. We therefore believe that our observations and conclusions obtained in our manuscript should not be affected by the occasional mixed wing morph.

Comment 4:- *It would be useful to clarify the role of sex in wing morph determination. From other BPH polyphenism papers, I understand there to be a small sex-effect, but it's not clear whether this is persistent for each genotype. I note that some experiments in this study focus only on females or males. It's not clear to me why that's the case. Some justification is necessary.*

Response to comment 4: Thank you for raising this issue. In brief, in our study, we conducted RNAi in both females and males, resulting in identical *morphological* phenotypes in both sexes. However, sexes may possibly differ in phenotype penetrance (see below). As a consequence, in this manuscript we report wing morph data derived from females in our main figures, and now provide the corresponding male data set in a supplementary figure (Supplementary Fig. 7), which was not included in the originally submitted manuscript.

The motivation for separating sexes in our analysis derives from earlier reports that suggested a sex bias in wing phenotypes following experimental manipulations. For instance, Dr. Xinda Li's group (China Jiliang University) reported that knockdown of *Akt* led to significant wing-morph ratio changes in females but not in males (Lin et al., 2016. *Int. J. Biol. Sci.* 12: 607-616). However, we found that knockdown of *Akt* resulted in almost 100% SW morphs in both females and males (Xu et al. 2015. *Nature* 519: 464-467), which obviously differs from data from Xinda Li's group. In another report also from Xinda Lin's group, knockdown of the gene encoding c-Jun NH2-terminal kinase (JNK) increased the proportion of SW females, but not males (Lin et al., *Insect Biochem. Mol. Biol.* 73: 55-61). We don't have any comments on this result since we have never repeated this experiment.

Our lab has been working on BPH wing dimorphism for more than 12 years. We never observed sex-biased wing morphs in all the experiments we have done except for knockdown of *tra-2*, a key gene of the sex determination pathway. Nymphal RNAi of *Tra-2* caused females to develop into infertile pseudomales containing undeveloped ovaries, while parental RNAi of *tra-2* resulted in 100% LW female offspring (Zhuo et al. 2017. *Genetics* 207: 1067-1078). We therefore think that outside of the sex

determination pathway sex biases in wing phenotypes are unlikely. However, to be certain we decided to analyse our results reported here one sex at a time.

Comment 5:- *The term "hierarchical regulator" used in the first paragraph and elsewhere in this manuscript seems like the unnecessary invention of a novel term. I would prefer to see *zfh1* referred to as "upstream" of FoxO and "parallel" to IIS, etc.*

Response to comment 5: In the revised manuscript, “hierarchical” has been replaced with “upstream”. Thank you.

Comment 6:- *Given the complexity of the dataset presented here, I do not wish to suggest more experiments. However, it does occur to me that their methods predicts that FoxO protein concentrations should differ between the nucleus and cytoplasm of each morph.*

Response to comment 6: In our previous report (Xu et al. 2015. *Nature* 519: 464-467), we investigated the nucleus and cytoplasm distribution of FoxO when the IIS pathway was perturbed. Activation of the IIS pathway increased phosphorylated level of FoxO, leading to a cytoplasmic accumulation of FoxO. By contrast, inactivation of the IIS pathway led to nucleus accumulation of dephosphorylated FoxO. Here, because knockdown of *Zfh1* decreased the FoxO protein level rather than affecting phosphorylation of FoxO, we did not examine sub-cellular location of FoxO.

Comment 7:- *I find the model presented in Fig 5 satisfying, and a good reflection of the data in this manuscript and other recent ones in their field. However, I would appreciate some speculation on upstream factors influencing *zfh1*. How might it be regulated? Is it likely to be environmentally regulated at all? If this is an evolutionarily conserved mechanisms, as they suggest, how or why might this point of regulation have arisen?*

Response to comment 7: Thank you for your kind words. Although various environmental cues, including crowding, host plant quality, photoperiod, and

temperature, have been reported to influence wing-morph switching, the dominant one is still missing (lines 70-73 in the Xu_manuscript_R1_clean copy). Besides, lab-raised SW-BPH populations are considered to be refractory to environmental cues that hypothetically stimulate LW development. This information was addressed in our previous review paper (Zhang et al. 2019. *Annu. Rev. Entomol.* 64: 297-314) and a recent paper from an independent group (Zhang et al., 2022. *Insect Science*. doi: 10.1111/1744-7917.13037). Therefore, we were unable to test how environmental cues affect the expression of *Zfh1*, and thus in the interest of length and focus of our ms decided not to speculate how *Zfh1* mediate environmental cues to regulate wing morphs. This work clearly remains to be done. For now the best we could do is to figure out how molecular signals inside the BPH body are transmitted to regulate this phenomenon.

Comment 8:

Specific comments

With reference to line numbers

- 12-13: "...escaping predators, and locating mates." Presumably, their arthropod ancestors could do those things too. Just cut this phrase.

Response to comment 9: We have revised the sentence (line 33-34 in the Xu_manuscript_R1_clean copy).

Comment 9:

- 15: "leading to" the logic here is questionable. Does flight loss precede brachyptery? It seems that cannot be stated with certainty. In any case, it's besides the point being made in this manuscript.

Response to comment 9: We have revised the sentence (line 34-36 in the Xu_manuscript_R1_clean copy).

Comment 10:

- 17: "...for studying the evolutionary significance of dispersal." I'm not sure statement is true actually. Roff and Harrisons theoretical work was foundational, but dispersal

ecology has, as far I can tell, not used polymorphic species as models. In any case, that's not the focus of this manuscript.

Response to comment 10: Two review papers published by Roff (1994) and Harrison (1980) were cited by this manuscript. Roff's paper introduced the evolution of flightlessness in insects and other animals, and Harrison's paper introduced how morphs are determined and how polymorphisms are maintained. In the revised manuscript, we have cited these two papers in line 35 and 44, respectively.

Comment 11:

- 19-20: Cut "hierarchical regulator"

Response to comment 11: The "hierarchical" has been replaced with "upstream".

Comment 12:

- 34: persuasive, not "persuading"

- 34-36: There is some grammatical issue here that makes the meaning of this sentence unclear.

Response to comment 12: We have revised the word and sentence.

Comment 13:

- 37: aphids, plural

Response to comment 13: We have revised the word.

Comment 14:

- 41: Publication in 2015 no longer seems "recent"!

Response to comment 14: We have deleted this word.

Comment 15:

- 70: rather than

Response to comment 15: We have revised the sentence.

Comment 16:

- 105: *grammar issue here*

Response to comment 16: We have corrected the sentence.

Comment 17:

- 124-5: *"zfh1 transcripts were equally distributed in T2 and T3 wing buds" -- The figure shows 1.7 fold differences actually. Admittedly this is much less than for Ubx. However, I would caution the authors against comparing the magnitude of differences here. The relative realtime PCR (using the delta-delta-Ct method) is comparative. The 100% on their y-axis could represent essentially zero transcripts, making such comparisons spurious.*

Response to comment 17: We indicated the significant differences of gene expression between T2W and T3 wing buds instead of fold changes in the revised manuscript (Fig. 6d, and lines 307-310).

Comment 18:

- 306: *Define IMW*

Response to comment 18: We defined the IMW in figure captions.

Comment 19:

- *Extended Data Fig 4: This is useful data, but perhaps present images more zoomed out to provide more anatomical context.*

Response to comment 19: In the revised manuscript, Extended Data Fig.4 has been changed into Fig.2. We used immunohistochemistry staining to show the anatomical structure of indirect flight muscles in a BPH body (Fig. 2a).

Comment 20:

- 596: *What software was used for read filtering?*

Response to comment 20: After Illumina sequencing, clean reads were generated by removing adapters, poly-N, and low-quality reads from the raw data using fastp algorithm (lines 362-364 in the Xu_manuscript_R1_clean copy).

REVIEWERS' COMMENTS

Reviewer #1 (Remarks to the Author):

The revised manuscript and the comments and response to reviewers has made this manuscript really stand out. It is well written, well organized, and the figures and captions are simple and clear. The important and significant results were there along with a compelling and comprehensive set of experiments and rationale but now they are clear and well justified. The work reported in this manuscript is a complicated story but it is an excellent and exciting new contribution. This paper will be an important contribution to the literature on phenotypic plasticity and will likely instigate new research into this important factor for the evolution of adaptation in animals.

Reviewer #2 (Remarks to the Author):

Authors have addressed all of my queries but comment #2 in my first review. A critical role of *zfh1* in myogenesis has been reported in *Drosophila* (Postigo et al., Mol. Cell. Biol. 1999). Short and long *zfh1* isoforms play an antagonistic function in maintenance and differentiation of IFM progenitors, respectively (Boukhatmi and Bray, eLife 2018). I think these relevant works should be cited and connectively discussed in appropriate contexts in this study, because these suggest that the *zfh1* function in the context of IFM development is not completely novel in planthoppers. To give an insight into the evolution of polyphenism in general, the authors could discuss what is the novelty of *zfh1* function to form the dimorphism of planthoppers. Remaining concerns that the authors might address are as below:

Line 121

A distal vein is clearly lost in the *zfh1* RNAi wing shown in Fig. 1g. Is it within a range of normal, or should it be "almost unchanged"?

Line 411

Please describe what is included in each sample; where was the cuticle sampled from? Does the leg include all three pairs? Which part of the gut was used?

Fig. 2

It is unclear which part of muscles is imaged with TEM. Please indicate it in Fig. 2a.

Fig. S3

Since a switch between long and short isoforms of *zfh1* is important for the differentiation of IFMs in *Drosophila*, please clearly mention the PCR amplified region (i.e. long or short isoform, or common) in the figure legend. To do that, please show whether there are conserved long and short isoforms in the plant hopper transcriptome.

Reviewer #3 (Remarks to the Author):

In my judgement, the revision adequately addresses criticisms raised by myself and the other reviewers.

Point-by-point response to referees

Comment #1: *Authors have addressed all of my queries but comment #2 in my first review. A critical role of *zfh1* in myogenesis has been reported in *Drosophila* (Postigo et al., Mol. Cell. Biol. 1999). Short and long *zfh1* isoforms play an antagonistic function in maintenance and differentiation of IFM progenitors, respectively (Boukhatmi and Bray, eLife 2018). I think these relevant works should be cited and connectively discussed in appropriate contexts in this study, because these suggest that the *zfh1* function in the context of IFM development is not completely novel in planthoppers. To give an insight into the evolution of polyphenism in general, the authors could discuss what is the novelty of *zfh1* function to form the dimorphism of planthoppers.*

Response to comment #1: Thank you for this suggestion. Several lines of study indicate that *Zfh1* functions in inhibiting muscle differentiation in *Drosophila* and vertebrates. In our study, we showed that knockdown of BPH *Zfh1* promoted IMF development. We provide this data to support that ds*Zfh1*^{LW} BPHs could fully resemble wild-type LW morphs.

In the revised manuscript, we cited four additional papers related to the *Zfh1* function reported previously, and the order of references changed accordingly. Given that the development of LW in BPHs mainly relies on cell proliferation, our findings indicate *Zfh1* functions in inhibiting wing cell proliferation in addition to myogenesis. This is a novel function of *Zfh1* not reported previously. We now provide a brief discussion section to address this.

31. Postigo, A. A. & Dean, D. C. ZEB, a vertebrate homolog of *Drosophila* *Zfh-1*, is a negative regulator of muscle differentiation. *EMBO J.* **16**, 3935-3943 (1997).
32. Postigo, A. A., Ward, E., Skeath, J. B. & Dean, D. C. *zfh-1*, the *Drosophila* homologue of ZEB, is a transcriptional repressor that regulates somatic myogenesis. *Mol. Cell. Biol.* **19**, 7255–7263 (1999).
33. Bray, S. & Boukhatmi, H. A population of adult satellite-like cells in *Drosophila* is maintained through a switch in RNA-isoforms. *eLife* **7**, e35954 (2018).
46. Gunage, R. D., Dhanyasi, N., Reichert, H. & VijayRaghavan, K. *Drosophila* adult muscle development and regeneration. *Semin. Cell Dev. Biol.* **72**, 56-66 (2017).

In *Drosophila*, two *Zfh1* isoforms (long and short) were identified and found to possess distinct functions in regulating myogenesis. To investigate whether BPH

encodes multiple *Zfh1* isoforms, we examined *Zfh1* transcripts using several transcriptomic databases generated through the second generation sequencing in our lab. However, only one *Zfh1* isoform was identified in BPHs. In the future, RNA-seq based on third generation sequencing is required to detect multiple BPH *Zfh1* isoforms if they do exist in BPHs. Currently, we do not know whether BPHs encode variant *Zfh1* isoforms with different functions or not.

Comment #2: Remaining concerns that the authors might address are as below:

Line 121

A distal vein is clearly lost in the zfh1 RNAi wing shown in Fig. 1g. Is it within a range of normal, or should it be “almost unchanged”?

Response to comment #2: The majority of wild-type BPHs contain this distal vein, however, a small fraction of wild-type BPHs do not. In this study, we found that the majority of *zfh1*-RNAi BPHs lost the distal vein. As such, the *zfh1*-RNAi BPH venation pattern is basically within a normal range. We adjusted the wording accordingly.

Comment #3: Line 411

Please describe what is included in each sample; where was the cuticle sampled from? Does the leg include all three pairs? Which part of the gut was used?

Response to comment #3: To investigate the spatial expression of *Zfh1*, fifth-instar Wt^{LW} and Wt^{SW} nymphs (n = 50) were collected for tissue dissection, and then total RNA was isolated from head, fat body, the abdominal cuticle, six legs, the whole digestive tract (gut), and nota (mesonotum and metanotum), respectively. We have revised the sentence in the methods section accordingly.

Comment #4: Fig. 2

It is unclear which part of muscles is imaged with TEM. Please indicate it in Fig. 2a.

Response to comment #4: We have updated Fig. 2a to indicate the muscle parts used for TEM.

Comment #5:Fig. S3

*Since a switch between long and short isoforms of *zfh1* is important for the differentiation of IFMs in *Drosophila*, please clearly mention the PCR amplified region (i.e. long or short isoform, or common) in the figure legend. To do that, please show whether there are conserved long and short isoforms in the plant hopper transcriptome.*

Response to comment #5: We indicated the PCR amplified region in Fig. 1b. To investigate whether BPH encodes multiple *Zfh1* isoforms, we examined *Zfh1* transcripts using several transcriptomic databases generated through the second generation sequencing in our lab. However, only one *Zfh1* isoform was identified in BPHs. In the future, RNA-seq based on third generation sequencing is required to detect multiple BPH *Zfh1* isoforms if they do exist in BPHs. Currently, we do not know whether BPHs encode variant *Zfh1* isoforms with different functions or not.